# FairWire: Fair Graph Generation

**O. Deniz Kose**
Department of Electrical Engineering and Computer Science
University of California Irvine
Irvine, CA, USA
okose@uci.edu

**Yanning Shen**[*]
Department of Electrical Engineering and Computer Science
University of California Irvine
Irvine, CA, USA
yannings@uci.edu

## Abstract

Machine learning over graphs has recently attracted growing attention due to its ability to analyze and learn complex relations within critical interconnected systems. However, the disparate impact that is amplified by the use of biased graph structures in these algorithms has raised significant concerns for their deployment in real-world decision systems. In addition, while synthetic graph generation has become pivotal for privacy and scalability considerations, the impact of generative learning algorithms on structural bias has not yet been investigated. Motivated by this, this work focuses on the analysis and mitigation of structural bias for both real and synthetic graphs. Specifically, we first theoretically analyze the sources of structural bias that result in disparity for the predictions of dyadic relations. To alleviate the identified bias factors, we design a novel fairness regularizer that offers a versatile use. Faced with the bias amplification in graph generation models brought to light in this work, we further propose a fair graph generation framework, FairWire, by leveraging our fair regularizer design in a generative model. Experimental results on real-world networks validate that the proposed tools herein deliver effective structural bias mitigation for both real and synthetic graphs.

## 1 Introduction

The volume of graph-structured data has been explosively growing due to the advancement in interconnected systems. In this context, machine learning (ML) over graphs attracts increasing attention (1), where specifically graph neural networks (GNNs) (2; 3; 4) have been proven to handle complex learning tasks over graphs, such as social recommendation (5), traffic flow forecasting (6).

Despite the increasing research focus on graph ML, the deployment of these algorithms in real-world decision systems requires guarantees preventing disparate impacts. Here, algorithmic disparity refers to the performance gap incurred by ML algorithms with respect to certain sensitive attributes protected by anti-discrimination laws or social norms (e.g., ethnicity, religion). While algorithmic bias is a concern over tabular data (7; 8), such bias becomes more critical for learning over graphs, as the use of graph structure in the algorithm design has been demonstrated to amplify the already existing bias (9). Motivated by this, in this work, we specifically focus on structural bias and consequently the disparity in the predictions of dyadic relationships among nodes. Note that since the link predictions

---

[*]corresponding author

38th Conference on Neural Information Processing Systems (NeurIPS 2024).

are informed by the proximity principle (nodes connect to other nodes that are similar to themselves), the bias in graph topology is directly reflected in link prediction. For example, in a social network, the denser connectivity within people from the same ethnic group leads to higher recommendation rates within these groups and may cause segregation in social relations (10). Hence, the development of a fair link prediction algorithm is of crucial importance to prevent potential segregation.

Fairness-aware algorithms typically require the knowledge of the sensitive attributes, the sharing of which can potentially create privacy concerns (11). From a scalability perspective, sharing real graphs is also accompanied by difficulties due to the ever-increasing size of graphs. All these factors contribute to the value of synthetic graph generation for a number of applications, such as recommendation systems (12), anomaly detection (13). For graph generation, data-driven models are shown to achieve state-of-the-art results (14; 15; 16), however, the fairness aspect of these models is under-explored. Recent works demonstrate that, in general, generative models tend to amplify the already existing bias in real data (17; 18), which is a potential issue for graph generation as well.

Faced with the aforementioned structural bias issues in graphs, in this work, we first carry out a theoretical analysis investigating the sources of such structural bias. Specifically, we deduce the factors that affect a commonly used bias metric, namely statistical parity (19), for link prediction. Guided by the theoretical findings, a novel fairness regularizer, $\mathcal{L}_{\text{FairWire}}$ is designed, which can be utilized for various graph-related problems, including link prediction and graph generation. In addition, an empirical analysis for a graph generation model is carried out, which reveals that the use of generative algorithms amplifies the already existing structural bias in real graph data. To resolve this issue, we design a new diffusion-based fair graph generation framework, FairWire, which leverages the proposed regularizer $\mathcal{L}_{\text{FairWire}}$. The training of diffusion model in FairWire is specifically designed to capture the correlations between the synthetic sensitive attributes and the graph connectivity, which enables fair model training with the existing techniques without revealing the real sensitive information. Overall, the contributions of this work can be summarized as follows:
**c1)** A theoretical analysis that reveals the causes of disparity in the predictions of dyadic relations between nodes is derived. Differing from the existing analyses regarding the statistical parity in link prediction, our analysis considers a more general setting where sensitive attributes can be non-binary.
**c2)** Based on the theoretical findings, we design a novel fairness regularizer, $\mathcal{L}_{\text{FairWire}}$, which can be directly utilized for link prediction, as well as for graph generation models to alleviate the structural bias in a task-agnostic way.
**c3)** We conduct an empirical analysis for the effect of graph generation models on the structural bias, which reveals the possible bias amplification related to these models.
**c4)** FairWire, a novel fair graph generation framework, is developed by leveraging $\mathcal{L}_{\text{FairWire}}$ within a diffusion model. The diffusion model is trained to capture the relations between the sensitive attributes and the graph topology, facilitating fair model training without private information leakage.
**c5)** Comprehensive experimental results over real-world networks show that the proposed framework can effectively mitigate structural bias and create fair synthetic graphs.

## 2 Related Work

**Fairness-aware learning over graphs.** Fairness-aware graph ML has attracted increasing attention in recent years (20; 21; 22). Existing works mainly focus on: 1) Group fairness (9; 23; 24; 25), 2) Individual fairness (26; 27), and 3) Counterfactual fairness (28; 29; 30). To mitigate bias in graph ML, different strategies are leveraged, including but not limited to adversarial regularization (9; 24; 31; 32), Bayesian debiasing (33), and graph editing (28; 34; 35; 36; 37). With a specific focus on link prediction, (19; 38) propose fairness-aware strategies to alter the adjacency matrix, while (39) designs a fairness-aware regularizer. Differing from the majority of existing strategies, our proposed design herein is guided and supported by theoretical results. Specifically, we rigorously analyze the factors in graph topology leading to disparity for link prediction considering non-binary sensitive attributes. Furthermore, the developed bias mitigation tool can be employed in a versatile manner for training the link prediction models, as well as for training the generative models to create synthetic, fair graphs.

**Synthetic Graph Generation.** Generating synthetic graphs that simulate the existing ones has been a topic of interest for a long time (40; 41), for which the success of deep neural networks has been demonstrated (12; 42; 43). Recently, the use of diffusion-based graph generative models has been increasing, due to their success in reflecting several important statistics of real graphs in the synthetic

ones (44; 45; 46; 16; 47; 48).To the best of our knowledge, the only existing work that considers fair graph generation is (49) which only outputs a graph structure without nodal features, sensitive attributes and node labels, and also requires class labels as input. Furthermore, it focuses on the disparities in generation quality for different sensitive groups as the fairness metric, which may not be predictive for the fairness performance in downstream tasks. In contrast, our scheme herein does not require any class labels or training of a particular downstream task. In addition, differing from the existing diffusion models, our generation of graph topology and nodal features is guided by the sensitive attributes, which enables us to capture the correlations between the synthetic sensitive attributes and synthetic graph structure/nodal features. To the best of our knowledge, this work provides the first fairness-aware diffusion-based graph generation framework.

## 3  Preliminaries

Given an input graph $\mathcal{G} := (\mathcal{V}, \mathcal{E})$, the focus of this study is investigating and mitigating the structural bias that may lead to unfair results for learning algorithms. Here, $\mathcal{V} := \{v_1, v_2, \cdots, v_N\}$ denotes the set of nodes and $\mathcal{E} \subseteq \mathcal{V} \times \mathcal{V}$ stands for the set of edges. Nodal features and the adjacency matrix of the input graph $\mathcal{G}$ are represented by $\mathbf{X} \in \mathbb{R}^{N \times F}$ and $\mathbf{A} \in \{0, 1\}^{N \times N}$, respectively, where $\mathbf{A}_{ij} = 1$ if and only if $(v_i, v_j) \in \mathcal{E}$. This work considers a single, potentially non-binary sensitive attribute for each node denoted by $\mathbf{s} \in \{1, \cdots, K\}^N$. In addition, $\mathbf{S} \in \{0, 1\}^{N \times K}$ represents the one-hot encoding of the sensitive attributes. For the graphs with class information, $\mathbf{y} \in \mathbb{R}^N$ denotes the class labels. Node representations output by the $l$th GNN layer are $\mathbf{H}^{l+1}$, with $\mathbf{h}_i^{l+1} \in \mathbb{R}^{F^{l+1}}$ denoting the learned hidden representations for node $v_i$. $\mathbf{x}_i \in \mathbb{R}^F$, and $s_i$ represent the feature vector, and the sensitive attribute of node $v_i$, respectively. Furthermore, $\mathcal{S}_k$ denotes the set of nodes whose sensitive attributes are equal to $k$. We define inter-edge set $\mathcal{E}^\chi := \{e_{ij} | v_i \in \mathcal{S}_a, v_j \in \mathcal{S}_b, a \neq b\}$, and intra-edge set $\mathcal{E}^\omega := \{e_{ij} | v_i \in \mathcal{S}_a, v_j \in \mathcal{S}_b, a = b\}$. Similarly, $d_i^\chi := \sum_{v_j \in \mathcal{V} - \mathcal{S}_a} A_{ij}, \forall v_i \in \mathcal{S}_a$ and $d_i^\omega := \sum_{v_j \in \mathcal{S}_a} A_{ij}, \forall v_i \in \mathcal{S}_a$ are the inter- and intra-degrees of node $v_i$, respectively. Finally, $U_\mathcal{A}$ represents the discrete uniform distribution over the elements of set $\mathcal{A}$.

## 4  Inspection and Mitigation of Structural Bias

This section first derives the conditions for a graph topology that leads to optimal statistical parity for link prediction. Guided by the obtained conditions, a fairness regularizer will then be presented. Statistical parity for link prediction is defined as $\Delta_{\text{SP}} := |\mathbb{E}_{(v_i, v_j) \sim U_\mathcal{V} \times U_\mathcal{V}}[g(v_i, v_j) \mid s_i = s_j] - \mathbb{E}_{(v_i, v_j) \sim U_\mathcal{V} \times U_\mathcal{V}}[g(v_i, v_j) \mid s_i \neq s_j]|$ (19), where $g(v_i, v_j)$ denotes the predicted probability for an edge between the nodes $i$ and $j$. To the best of our knowledge, our analysis is the first theoretical investigation for the relation between $\Delta_{\text{SP}}$ and the graph topology considering multi-valued sensitive attributes, thus it generalizes previous findings with binary sensitive attributes (19).

### 4.1  Bias Analysis

This subsection derives the conditions for a fair graph topology that achieves optimal statistical parity in the ensuing link prediction task. First, we will introduce the GNN model considered in this work.

**GNN model:** Throughout the analysis, a stochastic graph view, $\tilde{\mathbf{A}}$, is adopted, i.e., $\tilde{A}_{ij}$ denotes the probability of an edge between the nodes $v_i$ and $v_j$, and $\tilde{A}_{ij} = \tilde{A}_{ji}$. Let $\mathbf{Z}^{l+1}$ represent the aggregated representations by the $l$th GNN layer with $i$th row $\mathbb{E}_{\tilde{\mathbf{A}}}[\mathbf{z}_i^{l+1}] := \sum_{v_j \in \mathcal{V}} \tilde{A}_{ij} \mathbf{c}_j^{l+1}$, where $\mathbf{c}_i^{l+1} := \mathbf{W}^l \mathbf{h}_i^l$. Then, the hidden representation output by $l$th GNN layer for node $v_i$ can be written as $\mathbb{E}_{\tilde{\mathbf{A}}}[\mathbf{h}_i^{l+1}] = \sigma(\sum_{v_j \in \mathcal{V}} \tilde{A}_{ij} \mathbf{W}^l \mathbf{h}_j^l) = \sigma(\mathbb{E}_{\tilde{\mathbf{A}}}[\mathbf{z}_i^{l+1}])$, where $\mathbf{W}^l$ is the weight matrix and $\sigma(\cdot)$ is the non-linear activation employed in the $l$th GNN layer.

The following assumptions are made for Theorem 1 that will be presented in this subsection:
**A1:** $\|\mathbf{c}_i\|_\infty \leq \delta, \forall v_i \in \mathcal{V}$.
**A2:** $\frac{\mathbb{E}_{\tilde{\mathbf{A}}}[d_i^\omega]}{|\mathcal{S}_k|} \geq \frac{\mathbb{E}_{\tilde{\mathbf{A}}}[d_i^\chi]}{N - |\mathcal{S}_k|}, \forall v_i \in \mathcal{S}_k, \forall k \in \{1, \cdots, K\}$.
**A3:** $\sum_{v_i, v_j \in \mathcal{V}} \tilde{A}_{ij} \gg \mathbb{E}_{\tilde{\mathbf{A}}}[d_i^\chi], \forall v_i \in \mathcal{V}$.

These assumptions are naturally satisfied by most of the real-world graphs. Assumption **A1** implies that the representations, $\mathbf{c}_i$'s, are finite. For **A2**, note that most of the real-world social networks have considerably more intra-edges than inter-edges (50), i.e., $\mathbb{E}_{\tilde{\mathbf{A}}}[d_i^\omega] \geq \mathbb{E}_{\tilde{\mathbf{A}}}[d_i^\chi]$. Thus, unless $|\mathcal{S}_i\| \gg \|\mathcal{S}_j|, i \neq j$ (extremely unbalanced sensitive group sizes), **A2** holds. Finally, **A3** holds with high probability as $\mathbb{E}_{\tilde{\mathbf{A}}}[d_i^\chi] = \sum_{v_i \in \mathcal{S}_{s_i}, v_j \in \mathcal{V} - \mathcal{S}_{s_i}} \tilde{A}_{ij}$. We also demonstrate that these assumptions are valid for the real-world networks we are using in Appendix A in order to further justify them.

Building upon these assumptions, Theorem 1 reveals the factors leading to the disparity between the representations of different sensitive groups obtained at any GNN layer. Specifically, it upper bounds the term $\delta_k^{(l+1)} := \|\mathbb{E}_{\tilde{\mathbf{A}},v_i \sim U_{\mathcal{S}_k}}[\mathbf{h}_i^{l+1} \mid s_i = k] - \mathbb{E}_{\tilde{\mathbf{A}},v_i \sim U_{(\mathcal{V}-\mathcal{S}_k)}}[\mathbf{h}_i^{l+1} \mid s_i \neq k]\|_2$. The proof of the theorem is presented in Appendix B.

**Theorem 1.** *The disparity between the representations of nodes in a sensitive group $\mathcal{S}_k$ and the representations of the remaining nodes output by the lth GNN layer, $\delta_k^{(l+1)}$, can be upper bounded by:*

$$\delta_k^{(l+1)} \leq L\left(\delta\sqrt{F^{(l+1)}}\left(\left|\frac{p_k^\omega}{|\mathcal{S}_k|} - \frac{p_k^\chi}{N-|\mathcal{S}_k|}\right| + \left|\frac{\sum_{v_i,v_j \in \mathcal{V}} \tilde{A}_{ij} - p_k^\omega - 2p_k^\chi}{N-|\mathcal{S}_k|} - \frac{p_k^\chi}{|\mathcal{S}_k|}\right|\right) + 2\sqrt{N}\Delta_z\right), \tag{1}$$

*where $L$ is the Lipschitz constant of the activation function $\sigma(\cdot)$, $\left\|\mathbf{z}_i^{l+1} - \mathrm{mean}(\mathbf{z}_j^{l+1} \mid v_j \in \mathcal{V})\right\|_\infty \leq \Delta_z, \forall v_i \in \mathcal{V}$, and $p_k^\chi := \sum_{v_i \in \mathcal{S}_k, v_j \notin \mathcal{S}_k} \tilde{A}_{i,j}, p_k^\omega := \sum_{v_i \in \mathcal{S}_k, v_j \in \mathcal{S}_k} \tilde{A}_{i,j}$.*

Representation disparity resulting from the aforementioned GNN-based aggregation is examined and explained by Theorem 1. The commonly used fairness measures, such as statistical parity (19), are naturally a function of the representation disparity. Herein, we further investigate the said relation between the representation disparity and $\Delta_{\mathrm{SP}}$ mathematically. Specifically, for a link prediction model described by a function $g(v_i, v_j) := \mathbf{h}_i^\top \Sigma \mathbf{h}_j$, Proposition 1 directly upper bounds $\Delta_{\mathrm{SP}}$. Here, $\mathbf{h}_i$ denotes the representation for node $v_i$ that is employed for the link prediction task, i.e., the hidden representations in the final layer. The proof of Proposition 1 is presented in Appendix C.

**Proposition 1.** *For a link prediction model described by $g(v_i, v_j) := \mathbf{h}_i^\top \Sigma \mathbf{h}_j$, $\Delta_{\mathrm{SP}}$ can be upper bounded by:*

$$\Delta_{\mathrm{SP}} \leq \sum_{k=1}^{K} \frac{|\mathcal{S}_k|}{N} q \|\Sigma\|_2 \delta^{max}, \tag{2}$$

*where $\|\mathbf{h}_i\|_2 \leq q, \forall v_i, \delta^{max} := \max_k(\|\mathbb{E}_{\tilde{\mathbf{A}},v_i \sim U_{\mathcal{S}_k}}[\mathbf{h}_i \mid s_i = k] - \mathbb{E}_{\tilde{\mathbf{A}},v_j \sim U_{(\mathcal{V}-\mathcal{S}_k)}}[\mathbf{h}_j \mid s_j \neq k]\|_2)$.*

Combining the findings of Theorem 1 and Proposition 1, Corollary 1 further demonstrates the factors (including the topological ones) that affect the resulting statistical parity in the link prediction task.

**Corollary 1.** *For a link prediction model $g(v_i, v_j) := (\mathbf{h}_i^{L+1})^\top \Sigma \mathbf{h}_j^{L+1}$, where $\mathbf{h}_j^{L+1}$ is the representation created by Lth (final) GNN layer, $\Delta_{\mathrm{SP}}$ can be upper bounded by:*

$$\Delta_{\mathrm{SP}} \leq \sum_{k=1}^{K} \frac{|\mathcal{S}_k|}{N} q \|\Sigma\|_2 L\left(\delta\sqrt{F^{(L+1)}}(\alpha_1 + \alpha_2) + 2\sqrt{N}\Delta_z\right),$$

*where $\alpha_1 := \left|\frac{p_k^\omega}{|\mathcal{S}_k|} - \frac{p_k^\chi}{N-|\mathcal{S}_k|}\right|$ and $\alpha_2 := \left|\frac{\sum_{v_i,v_j \in \mathcal{V}} \tilde{A}_{ij} - p_k^\omega - 2p_k^\chi}{N-|\mathcal{S}_k|} - \frac{p_k^\chi}{|\mathcal{S}_k|}\right|$.*

## 4.2 A Regularizer for Fair Connections

The bias analysis in Subsection 4.1 brings to light the factors resulting in topological bias for a probabilistic graph connectivity. Corollary 1 shows that the topological bias can be minimized if $\alpha_1 = 0$ and $\alpha_2 = 0$. One can obtain $\alpha_1 = 0$ by ensuring $\frac{p_k^\omega}{p_k^\chi} = \frac{|\mathcal{S}_k|}{N-|\mathcal{S}_k|}, \forall k$. Meanwhile, $\alpha_2 = 0$ if $p_k^\omega = \sum_{v_i,v_j \in \mathcal{V}}(\tilde{\mathbf{A}}) - c|\mathcal{S}_k| - cN$ and $p_k^\chi = c|\mathcal{S}_k|$ for any constant $c \in \mathbb{R}$. Overall, the optimal values of $p_k^\omega$ and $p_k^\chi$ that minimize both $\alpha_1$ and $\alpha_2$ follow as $(p_k^\omega)^* = \frac{\sum_{v_i,v_j \in \mathcal{V}} \tilde{A}_{ij}|\mathcal{S}_k|^2}{N^2}$ and $(p_k^\chi)^* = \frac{(\sum_{v_i,v_j \in \mathcal{V}} \tilde{A}_{ij})(N|\mathcal{S}_k|-|\mathcal{S}_k|^2)}{N^2}$. Therefore, in order to mitigate structural bias, we can design a regularizer that pushes the expected number of inter-edges and intra-edges towards $(p_k^\chi)^*$ and $(p_k^\omega)^*$:

$\mathcal{L} := \sum_{k=1}^{K} |\sum_{v_i,v_j \in \mathcal{V}}(\tilde{\mathbf{A}} \odot (\mathbf{Se}_k)(\mathbf{Se}_k)^\top)_{i,j} - (p_k^\omega)^*| + |\sum_{v_i,v_j \in \mathcal{V}}(\tilde{\mathbf{A}} \odot (\mathbf{Se}_k)(\mathbf{1}-(\mathbf{Se}_k))^\top)_{i,j} -$

$(p_k^\chi)^*|$. Here, $\mathbf{e}_k \in \mathbb{R}^K$ is the basis vector with only non-zero entry 1, at the $k$th element, and $\odot$ denotes the Hadamard product. Note that such a regularizer is compatible with any learning algorithm that outputs probabilities of all possible edges in the graph, e.g., topology inference algorithms.

Although $\mathcal{L}$ can be applied to several graph ML algorithms and its theory-guided design can promise effective topological bias mitigation, its design requires a single-batch learning setting due to the definitions of $p_k^\chi$ and $p_k^\omega$ (resulting in a complexity growing exponentially with $N$). Specifically, $p_k^\chi$ and $p_k^\omega$ are calculated based on all edge probabilities related to the all nodes in $\mathcal{S}_k$. Therefore, regularizing the values of $p_k^\chi$ and $p_k^\omega$ will lead to scalability issues for large graphs. To tackle this challenge, we only focus on the optimal ratio between the expected number of intra- and inter-edges, i.e., $\frac{p_k^\omega}{p_k^\chi} = \frac{|\mathcal{S}_k|}{N - |\mathcal{S}_k|}, \forall k \in \{1, \cdots, K\}$, which is governed by $\alpha_1$. The idea is to manipulate the ratio between the expected number of intra- and inter-edges in each mini-batch of nodes for a better scalability. We call the corresponding batch-wise fairness regularizer $\mathcal{L}_{\text{FairWire}}$, which follows as

$$\mathcal{L}_{\text{FairWire}}(\tilde{\mathbf{A}}, \mathcal{B}) := \sum_{k=0}^{K} \left| \frac{\sum_{v_i, v_j \in \mathcal{B}} (\tilde{\mathbf{A}} \odot (\mathbf{S}\mathbf{e}_k)(\mathbf{S}\mathbf{e}_k)^\top)_{ij}}{|\mathcal{S}_k|} - \frac{\sum_{v_i, v_j \in \mathcal{B}} (\tilde{\mathbf{A}} \odot (\mathbf{S}\mathbf{e}_k)(\mathbf{1} - (\mathbf{S}\mathbf{e}_k))^\top)_{ij}}{N - |\mathcal{S}_k|} \right|, \tag{3}$$

where $\mathcal{B}$ denotes the set of nodes within the utilized minibatch. Note that the aforementioned versatile use of $\mathcal{L}$ also applies to $\mathcal{L}_{\text{FairWire}}$, which can directly be used in topology inference tasks. Specifically, for link prediction, the following loss function can be employed in training to combat bias:

$$\mathcal{L}_{lp} = \sum_{v_i, v_j \in \mathcal{B}} \mathcal{L}_{CE}(\tilde{\mathbf{A}}_{ij}, \mathbf{A}_{ij}) + \lambda \mathcal{L}_{\text{FairWire}}(\tilde{\mathbf{A}}, \mathcal{B}), \tag{4}$$

where $\tilde{\mathbf{A}}_{ij}$ denotes the predicted probability by the algorithm for an edge between $v_i$ and $v_j$, and $\mathcal{L}_{CE}$ is cross-entropy loss. The hyperparameter $\lambda$ is used to adjust the weight of fairness in training.

## 5 Fair Graph Generation

Generating synthetic graphs that capture the structural characteristics in real data attracts increasing attention as a promising remedy for scalability (ever-increasing size of real-world graphs) and privacy issues. Especially, sharing real sensitive attributes for fair model training exacerbates the privacy concerns due to the sensitive attribute leakage problem (11). Thus, creating synthetic graphs with generative models becomes instrumental in applications over interconnected systems. In this work, we focus on diffusion models whose success in capturing the original data distribution has been shown for various types of networks (45; 46; 16; 47). Despite the growing interest in these models, their effects on fairness have not yet been investigated, which limits their use in critical real-world decision systems. Motivated by this, in Subsection 5.1, we first empirically analyze the impact of diffusion models on the algorithmic bias by comparing the original and synthetic graphs in terms of different fairness metrics for link prediction. This empirical investigation reveals that the algorithmic bias is amplified while using generative models for graph creation. To resolve this critical issue, we develop FairWire in Subsection 5.2, a fair graph generation framework, which leverages our proposed regularizer $\mathcal{L}_{\text{FairWire}}$ during the training of a diffusion model.

### 5.1 Diffusion Models and Structural Bias

To evaluate the effect of synthetic graph generation on bias, we first sample 10 different synthetic graphs for each of the 4 real-world networks (see Table 7 in Appendix E and Subsection 6.1 for more details on the datasets). Synthetic graphs are sampled using a diffusion model that is trained following the setup in (47), which is a state-of-the-art algorithm for diffusion-based graph generation. Upon creating graphs, we evaluate them for the link prediction task on the same test set (generated from the real data) and report the corresponding utility (AUC) and fairness performance. Fairness performance is measured via two widely used bias metrics, statistical parity ($\Delta_{\text{SP}}$) and equal opportunity ($\Delta_{\text{EO}}$) (19) for which lower values indicate better fairness (see Subsection 6.1 for more details on the link prediction model and evaluation metrics). The obtained results are presented in Table 1.

In Table 1, $\mathcal{G}$ denotes the original graphs, and the synthetic graphs are represented by $\tilde{\mathcal{G}}$. Overall, Table 1 shows that graph generation via diffusion models indeed amplifies the already existing bias in the original graphs consistently for all the considered datasets. This brings the potential bias-related issues in synthetic graph creation to light and calls for robust bias mitigation solutions.

Table 1: Comparative results

| | Cora | | | Citeseer | | |
|---|---|---|---|---|---|---|
| | Accuracy (%) | $\Delta_{SP}$ (%) | $\Delta_{EO}$ (%) | Accuracy (%) | $\Delta_{SP}$ (%) | $\Delta_{EO}$(%) |
| $\mathcal{G}$ | 94.92 | 27.71 | 11.53 | 95.76 | 29.05 | 9.53 |
| $\tilde{\mathcal{G}}$ | $87.29 \pm 1.09$ | $35.72 \pm 1.74$ | $13.27 \pm 0.81$ | $92.19 \pm 1.06$ | $37.56 \pm 1.29$ | $13.52 \pm 0.92$ |
| | Amazon Photo | | | Amazon Computer | | |
| | Accuracy (%) | $\Delta_{SP}$ (%) | $\Delta_{EO}$ (%) | Accuracy (%) | $\Delta_{SP}$ (%) | $\Delta_{EO}$(%) |
| $\mathcal{G}$ | 96.91 | 32.58 | 8.24 | 96.14 | 22.90 | 4.63 |
| $\tilde{\mathcal{G}}$ | $94.45 \pm 0.21$ | $33.49 \pm 0.28$ | $10.01 \pm 0.56$ | $94.04 \pm 0.26$ | $23.56 \pm 0.55$ | $6.23 \pm 0.49$ |

## 5.2 FairWire: A Fair Graph Generation Framework

The proposed fairness regularizer in Subsection 4.2, $\mathcal{L}_{\text{FairWire}}$, can be utilized in two different settings: $i$) during model training for link prediction, $ii$) for training a graph generation model in a task-agnostic way. Note that for both cases, a model is trained to predict a probabilistic graph adjacency matrix, $\tilde{\mathbf{A}}$, upon which $\mathcal{L}_{\text{FairWire}}$ can be employed. Both use cases can facilitate several fairness-aware graph-based applications. That said, the bias amplification issue in generative models (also observed from Table 1) makes creating fair graphs via graph generation models of particular interest.

The proposed fair graph generation framework, FairWire, is built upon structured denoising diffusion models for discrete data (51). In the forward diffusion, FairWire employs a Markov process to create noisy graph data samples by independently adding or deleting edges. For denoising, a message-passing neural network (MPNN) is trained to predict the clean graph based on noisy samples by using the guidance of sensitive attributes (and node labels if available). Finally, we sample synthetic graphs with the guidance of synthetic sensitive attributes that are initialized based on their distribution in the original data. If input graph has also node labels, during graph generation, these node labels are sampled based on their distribution conditioned on the sensitive attributes in the original graph. In the sequel, as our main novelty lies in the denoising process, we discuss the training process of FairWire (reverse diffusion process) in more detail, while the forward diffusion and sampling processes are explained in Appendix D. Note that the diffusion process is presented for attributed graphs, where synthetic nodal features $\tilde{\mathbf{X}}$ are also generated. However, the proposed approach can be readily adapted to graphs without nodal features.

**Reverse diffusion process:** For denoising, we train an MPNN, $\phi_\theta$ parametrized by $\theta$, which is shown to be a scalable solution for the generation of large, attributed graphs (47). Specifically, $\phi_\theta$ inputs a noisy version of the input graph and the original sensitive attributes described by $\mathbf{X}^t, \mathbf{A}^t, \mathbf{S}$ and aims to recover the original nodal features $\mathbf{X}^0$ and graph topology $\mathbf{A}^0$. Here, $\mathbf{A}^0 \in \mathbb{R}^{N \times N \times 2}$ denotes the one-hot representations for the edge labels. Note that the sensitive attributes are used to guide the MPNN to capture the relations between them and graph topology. Therefore, the sensitive attributes are initialized and kept the same during both training and sampling (the original distribution of sensitive attributes is used to initialize them during sampling). For a node $v$, the message passing at the $l$th layer can be described as:

$$\mathbf{h}_v^{(t,l+1)} = \sigma\left(\mathbf{W}_{T \to H}^{(l)}\mathbf{h}_t + \mathbf{b}_H^{(l)} + \sum_{u \in \mathcal{N}^{(t)}(v)} \frac{1}{\left|\mathcal{N}^{(t)}(v)\right|}\left[\mathbf{h}_u^{(t,l)}\|\mathbf{S}_u^{(l)}\right]\mathbf{W}_{[H,S] \to H}^{(l)}\right), \quad (5)$$

$$\mathbf{S}_v^{(l+1)} = \sigma\left(\mathbf{b}_S^{(l)} + \sum_{u \in \mathcal{N}^{(t)}(v)} \frac{1}{\left|\mathcal{N}^{(t)}(v)\right|}\mathbf{S}_u^{(l)}\mathbf{W}_{S \to S}^{(l)}\right), \quad (6)$$

where $\|$ stands for the concatenation operator. In this aggregation, $\mathbf{W}_{T \to H}^{(l)}, \mathbf{W}_{[H,S] \to H}^{(l)}, \mathbf{W}_{S \to S}^{(l)}, \mathbf{b}_H^{(l)}$ and $\mathbf{b}_S^{(l)}$ are all learnable parameters, while $\sigma(\cdot)$ consists of ReLU (52) and LayerNorm (53) layers. In addition, $\mathbf{H}^{(t,0)}$ and $\mathbf{h}_t$ are initialized as hidden representations created for $\mathbf{X}^t$ and time step $t$ via multi-layer perceptrons (MLP), respectively, and $\mathbf{S}^{(0)} = \mathbf{S}$. After creating hidden representations for nodes and their sensitive attributes, final representation for a node $v$ is generated via $\mathbf{h}_v = \mathbf{h}_v^{(t,0)}\|\mathbf{h}_v^{(t,1)}\| \cdots \|\mathbf{S}_v^{(0)}\|\mathbf{S}_v^{(1)}\| \cdots \|\mathbf{h}_t$. Note that when node labels are available, their one-hot representations $\mathbf{Y}$, are also employed in this MPNN in the same way as $\mathbf{S}$ are utilized. Based on these final representations, node attributes, and edge labels are predicted. To create fair graph connections in the synthetic graphs, we regularize the predicted edge probabilities, $\tilde{\mathbf{A}}$, via the designed fairness

regularizer $\mathcal{L}_{\text{FairWire}}$. Overall, the training loss of the MPNN follows as:

$$\sum\nolimits_{v_i \in \mathcal{B}} \mathcal{L}_{CE}\big(\tilde{\mathbf{X}}_{i:}, \mathbf{X}^0_{i:}\big) + \sum\nolimits_{v_i, v_j \in \mathcal{B}} \mathcal{L}_{CE}\big(\tilde{\mathbf{A}}_{ij:}, \mathbf{A}^0_{ij:}\big) + \lambda \mathcal{L}_{\text{FairWire}}(\tilde{\mathbf{A}}, \mathcal{B}), \qquad (7)$$

where $\lambda$ adjusts the focus on the fairness regularizer.

**Remark (Applicability to general generative models):** Although the designed regularizer in Subsection 4.2 is embodied in a diffusion-based graph generation framework in Subsection 5.2, $\mathcal{L}_{\text{FairWire}}$ can be utilized in any generative model outputting synthetic graph topologies as a fairness regularizer on the connections, including but not limited to graph autoencoder-based or random walk-based graph generation models.

**Remark (Creation of synthetic sensitive attributes):** We design a generative framework in Subsection 5.2 that outputs synthetic sensitive attributes whose effect on the connections is reflected by inputting them in the training of MPNN. We emphasize that the creation of these synthetic sensitive attributes also enables the use of existing fairness-aware schemes on the created graphs without leaking the real sensitive attributes.

## 6  Experiments

### 6.1  Datasets and Experimental Setup

**Datasets.** In the experiments, four attributed networks are employed, namely Cora, Citeseer, Amazon Photo and Amazon Computer for link prediction. Cora and Citeseer are widely utilized citation networks, where the articles are nodes and the network topology depicts the citation relationships between these articles (54). Amazon Photo and Amazon Computer are product co-purchase networks, where the nodes are the products and the links are created if two products are often bought together (55). In addition to link prediction, we also evaluate the synthetic graphs on node classification, where the German credit (56) and Pokec-n (9) graphs are employed. For more details on the datasets and their statistics, please see Appendix E.

**Experimental Setup.** In this section, we first report the performance of $\mathcal{L}_{\text{FairWire}}$ for link prediction. For this task, the area under the curve (AUC) is employed as the utility metric. As fairness metrics, statistical parity and equal opportunity definitions in (19; 37) are used, where $\Delta_{\text{SP}} := |\mathbb{E}_{(v_i,v_j) \sim U_\mathcal{V} \times U_\mathcal{V}}[\tilde{A}_{ij} = 1 \mid s_i = s_j] - \mathbb{E}_{(v_i,v_j) \sim U_\mathcal{V} \times U_\mathcal{V}}[\tilde{A}_{ij} = 1 \mid s_i \neq s_j]|$ and $\Delta_{\text{EO}} := |\mathbb{E}_{(v_i,v_j) \sim U_\mathcal{V} \times U_\mathcal{V}}[\tilde{A}_{ij} = 1 \mid A_{ij} = 1, s_i = s_j] - \mathbb{E}_{(v_i,v_j) \sim U_\mathcal{V} \times U_\mathcal{V}}[\tilde{A}_{ij} = 1 \mid A_{ij} = 1 s_i, \neq s_j]|$. Lower values for $\Delta_{\text{SP}}$ and $\Delta_{\text{EO}}$ indicate better fairness performance.

To evaluate the generated synthetic graphs, we use both the link prediction and node classification tasks. Herein, we sample 10 synthetic graphs for each dataset with the trained diffusion models. Afterward, we train link prediction/node classification models (for more details on these models, please see Appendix G) on the sampled graphs, and test these models on the real graphs $\mathcal{G}$ (the test set is the same for all baselines and FairWire). Here, we consider the scenario where there is no access to the real graphs due to privacy concerns, and the models are trained on the synthetic graphs for downstream tasks. To evaluate these synthetic graphs on link prediction, the same utility and fairness metrics as in the link prediction task are used. For node classification, accuracy is employed as the utility measure with $\Delta_{SP} := |P(\hat{y}_j = 1 \mid s_j = 0) - P(\hat{y}_j = 1 \mid s_j = 1)|$, and $\Delta_{EO} := |P(\hat{y}_j = 1 \mid y_j = 1, s_j = 0) - P(\hat{y}_j = 1 \mid y_j = 1, s_j = 1)|$ being the fairness metrics.

For more details on the training of link prediction, node classification, diffusion models, and the hyperparameter selection for FairWire and baselines, see Appendix G. A sensitivity analysis is also provided in Appendix H for the effect of hyperparameter $\lambda$ in (4) and in (7). Note that the performance of the generative algorithms is generally reported in terms of the distances between the statistics of real data and the synthetic ones, instead of the fairness performance. For completeness, we report the distance metrics for node degree distribution and clustering coefficient distribution in Appendix F.

**Baselines.** For link prediction, fairness-aware baselines include adversarial regularization (9), FairDrop (37), and FairAdj (19). For graph generation, FairGen (49), is the only existing fairness-aware baseline designed for node classification. For a comprehensive evaluation, we also employ adversarial regularization (9) and FairAdj (19) as in-processing and post-processing fairness-aware strategies within the generative model. For more details on the baselines, please see Appendix G.

Table 2: Comparative link prediction results.

| | Cora | | | Citeseer | | |
|---|---|---|---|---|---|---|
| | AUC (%) | $\Delta_{SP}$ (%) | $\Delta_{EO}$ (%) | AUC (%) | $\Delta_{SP}$ (%) | $\Delta_{EO}$(%) |
| GNN | **94.43 ± 0.74** | 27.01 ± 1.38 | 9.11 ± 1.43 | **96.16 ± 0.28** | 27.40 ± 1.24 | 7.37 ± 1.33 |
| Adversarial | 87.77 ± 1.64 | 24.33 ± 6.36 | 2.96 ± 2.24 | 93.53 ± 0.51 | 13.97 ± 10.36 | 6.52 ± 4.68 |
| FairDrop | 94.10 ± 0.81 | 7.86 ± 4.30 | 4.05 ± 0.32 | 95.92 ± 0.42 | 13.77 ± 6.15 | 5.60 ± 1.85 |
| FairAdj | 82.01 ± 1.56 | 14.76 ± 0.89 | 7.35 ± 1.24 | 84.76 ± 1.08 | 16.00 ± 11.93 | 5.91 ± 3.42 |
| $\mathcal{L}_{\text{FairWire}}$ | 92.18 ± 1.03 | **4.76 ± 0.24** | **2.05 ± 0.37** | 96.00 ± 0.23 | **8.62 ± 0.80** | **1.29 ± 0.68** |
| | Amazon Photo | | | Amazon Computer | | |
| | AUC (%) | $\Delta_{SP}$ (%) | $\Delta_{EO}$ (%) | AUC (%) | $\Delta_{SP}$ (%) | $\Delta_{EO}$(%) |
| GNN | 97.01 ± 0.26 | 32.65 ± 0.95 | 8.01 ± 0.52 | **96.13 ± 0.06** | 23.70 ± 0.79 | 5.51 ± 0.79 |
| Adversarial | 96.17 ± 0.09 | 29.57 ± 0.91 | 8.03 ± 1.05 | 95.64 ± 0.12 | 22.72 ± 1.11 | 4.71 ± 0.99 |
| FairDrop | 95.36 ± 0.33 | 28.63 ± 1.39 | 8.49 ± 1.54 | 95.61 ± 0.13 | 21.30 ± 0.59 | 4.30 ± 0.77 |
| $\mathcal{L}_{\text{FairWire}}(\lambda = c)$ | **97.25 ± 0.11** | **27.75 ± 0.52** | **7.11 ± 0.41** | 96.05 ± 0.05 | **20.44 ± 0.44** | **4.24 ± 0.51** |
| $\mathcal{L}_{\text{FairWire}}(\lambda = 5c)$ | 94.85 ± 0.32 | **24.61 ± 0.96** | **6.24 ± 1.22** | 93.86 ± 0.23 | **15.36 ± 1.02** | **0.17 ± 0.21** |

Table 3: Comparative results for graph generation on Link Prediction.

| | Cora | | | Citeseer | | |
|---|---|---|---|---|---|---|
| | AUC (%) | $\Delta_{SP}$ (%) | $\Delta_{EO}$ (%) | AUC (%) | $\Delta_{SP}$ (%) | $\Delta_{EO}$(%) |
| $\mathcal{G}$ | 94.92 | 27.71 | 11.53 | 95.76 | 29.05 | 9.53 |
| $\tilde{\mathcal{G}}$ | **87.29 ± 1.09** | 35.72 ± 1.74 | 13.27 ± 0.81 | **92.19 ± 1.06** | 37.56 ± 1.29 | 13.52 ± 0.92 |
| FairAdj | 82.13 ± 1.07 | 15.47 ± 2.39 | 6.26 ± 2.05 | 82.67 ± 2.78 | **15.45 ± 2.68** | 7.98 ± 1.47 |
| Adversarial | 83.66 ± 5.64 | 16.35 ± 9.80 | 7.82 ± 5.84 | 89.59 ± 2.70 | 24.20 ± 5.82 | 10.34 ± 1.66 |
| FairWire | 86.49 ± 2.79 | **12.91 ± 6.35** | **4.31 ± 3.59** | 91.27 ± 2.78 | 18.35 ± 6.91 | **7.80 ± 2.76** |

## 6.2 Link Prediction Results

Comparative results for the link prediction task are presented in Table 2, where we consider the setting $\mathcal{L}_{\text{FairWire}}$ is employed as a fairness regularizer while training a GNN model for link prediction. The natural baseline here is to employ the same GNN model without any fairness interventions, which is denoted by GNN in Table 2. Note that in Table 2, $c$ equals to $0.01$ for Amazon Photo and the results are presented for $c = 0.1$ for Amazon Computer.

The results in Table 2 demonstrate that employing $\mathcal{L}_{\text{FairWire}}$ as a fairness regularizer leads to better fairness measures compared to the naive baseline, while also providing similar utility. Specifically, the proposed regularizer is observed to improve both fairness metrics, $\Delta_{SP}$ and $\Delta_{EO}$, with improvements ranging from $20\%$ to $80\%$ for every evaluated dataset compared to the natural baseline, GNN. Furthermore, the obtained results show that $\mathcal{L}_{\text{FairWire}}$ also outperforms the fairness-aware baselines Adversarial (9), FairDrop (37), and FairAdj (19) in both fairness metrics. For certain datasets (e.g., Amazon Photo, Amazon Computer), we report the results of FairWire for different values of $\lambda$ to illustrate the trade-off between fairness and utility and to show that FairWire leads to a better trade-off compared to the other fairness-aware baselines. Note that we could not include the results of FairAdj over the product networks (i.e., Amazon Photo, Amazon Computer) due to its substantial memory use during its optimization process, which led to out-of-memory errors for the infrastructure we use. In addition to the improved fairness performance, it can be observed that the employment of $\mathcal{L}_{\text{FairWire}}$ generally results in the lowest standard deviation values for fairness metrics, which demonstrates the stability of the proposed strategy for bias mitigation. Overall, the results corroborate the effectiveness of $\mathcal{L}_{\text{FairWire}}$ in enhancing fairness while also providing similar utility compared to the state-of-the-art fairness-aware baselines.

## 6.3 Results for Graph Generation

Comparative results for graph generation are presented in Tables 3 and 4, where the link prediction and node classification tasks are used to evaluate the synthetic graphs, respectively. In these tables, $\mathcal{G}$ represents the original data, and $\tilde{\mathcal{G}}$ stands for the synthetic graphs generated by the fairness-agnostic GraphMaker (47). Overall, Tables 3 and 4 show that FairWire improves fairness metrics compared to $\tilde{\mathcal{G}}$, fairness-agnostic synthetic graphs created via diffusion, without a significant utility loss for

Table 4: Comparative results for graph generation on Node Classification.

| | German | | | Pokec-n | | |
|---|---|---|---|---|---|---|
| | Acc (%) | $\Delta_{SP}$ (%) | $\Delta_{EO}$ (%) | Acc (%) | $\Delta_{SP}$ (%) | $\Delta_{EO}$ (%) |
| $\mathcal{G}$ | 70.00 | 2.13 | 1.78 | 68.73 | 8.58 | 9.68 |
| FairGen | **73.60** | 28.71 | 15.34 | 51.73 | 0.00 | 0.00 |
| $\tilde{\mathcal{G}}$ | $68.92 \pm 2.37$ | $2.61 \pm 5.83$ | $2.29 \pm 5.06$ | $66.19 \pm 2.05$ | $3.63 \pm 2.58$ | $2.66 \pm 2.50$ |
| FairAdj | $70.08 \pm 1.08$ | $2.17 \pm 4.49$ | $1.11 \pm 2.24$ | - | - | - |
| Adversarial | $70.00 \pm 0.62$ | $1.57 \pm 2.70$ | $1.34 \pm 2.86$ | **69.36** $\pm 0.70$ | $2.16 \pm 1.73$ | $2.73 \pm 2.01$ |
| FairWire | $69.76 \pm 0.51$ | **0.63** $\pm 1.53$ | **0.30** $\pm 0.61$ | $68.23 \pm 0.45$ | **1.91** $\pm 0.92$ | **1.35** $\pm 0.92$ |

both link prediction and node classification. Specifically, FairWire can achieve improvements in both $\Delta_{SP}$ and $\Delta_{EO}$ ranging from 25% to 90% for all datasets compared to $\tilde{\mathcal{G}}$ with similar utility.

Note that, for link prediction, the fairness improvement reported for FairAdj in Table 3 is accompanied by a significant utility drop. Specifically, for a larger $\lambda$ value (i.e., $\lambda = 1$), FairWire can provide better fairness measures ($\Delta_{SP} = 7.01 \pm 6.25$ and $\Delta_{EO} = 3.06 \pm 2.95$) on Citeseer with a similar accuracy ($83.47 \pm 7.79$) to FairAdj. Thus, the results in Table 3 demonstrate that FairWire provides a better utility/fairness trade-off compared to fairness-aware baselines on all evaluated datasets.

In Table 4, similar to the link prediction experiments (Table 2), the results of FairAdj for the Pokec-n network could not be obtained due to computational limitations. For FairGen (49), we directly input the synthetic graph output by the algorithm to the node classification model we train, thus the results are obtained for a single synthetic graph. A possible explanation for the better accuracy of FairGen on the German dataset is that the algorithm is observed to output a denser synthetic network, which might be useful for the utility. It is observed that the synthetic graph output by FairGen for Pokec-n was not informative enough for the node classification task (we provide the codes for the FairGen algorithm in our supplementary material for the reproducibility of these results.) All in all, the results in Table 4 signify that the superior performance of FairWire in terms of fairness/utility trade-off can also be observed for node classification, which validates the efficacy of FairWire in creating fair synthetic graphs that also capture the real data distribution.

## 6.4 Visualization of Synthetic Graphs

Our analysis in Subsection 4.1 reveals that the ratio of intra- (edges connecting the same sensitive group) and inter-edges (edges between different sensitive groups) is a factor contributing to the structural bias. Specifically, the bias factor $\alpha_1$ is minimized when $\frac{p_k^\omega}{p_k^\chi} = \frac{|\mathcal{S}_k|}{N - |\mathcal{S}_k|}, \forall k$, where $p_k^\omega$ and $p_k^\chi$ are the expected number of intra- and inter-edges for the nodes in $\mathcal{S}_k$. This finding suggests that for a graph with multiple ($> 2$) sensitive groups, given the sizes of sensitive groups are not catastrophically unbalanced, the number of inter-edges (related to $p_k^\chi$) should be larger than the number of intra-edges (related to $p_k^\omega$) to alleviate

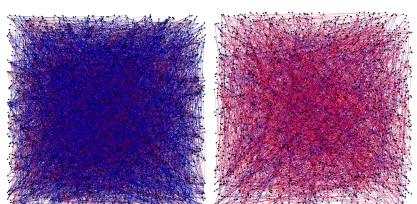

Figure 1: Distribution of the intra-edges (blue) and inter-edges (red) in the synthetic graphs created for Cora dataset by Graph-Maker (47) (left) and FairWire (right).

structural bias (i.e., $|\mathcal{S}_k| \leq N - |\mathcal{S}_k|$). However, for graphs encountered in several domains, the number of intra-edges is significantly larger than the number of inter-edges, due to the homophily principle (10). Motivated by this, in Figure 1, we visualize the distributions of intra- and inter-edges in synthetic graphs created by i) a fairness-agnostic strategy, GraphMaker (47), and ii) FairWire, for Cora. In Figure 1, intra- and inter-edges are colored with blue and red, respectively. Figure 1 reveals that the graph created by GraphMaker (47) predominantly consists of intra-edges, leading to the structural bias reflected in Table 3. In contrast, FairWire exhibits a remarkable balancing effect, which provides a potential explanation for the improvement in fairness.

# 7 Conclusion

This study focuses on the investigation and mitigation of structural bias for both real and synthetic graphs, where a novel fairness regularizer, $\mathcal{L}_{\text{FairWire}}$, is designed to alleviate the effects of bias factors identified in a developed theoretical bias analysis. Furthermore, the proposed fairness regularizer is leveraged in a fair graph generation framework, FairWire, which alleviates the bias amplification observed in graph generative models. Experimental results corroborate the effectiveness of the proposed tools in bias mitigation for both real and synthetic graphs.

**Limitations:** This paper considers the setting where sensitive attributes are available during model training, which might limit its use for certain real-world applications. Thus, one future direction of this work would be to consider the partial availability of these sensitive attributes in the input graph data. Furthermore, although we showed that real-world graphs typically satisfy the assumptions in Subsection 4.1, another possible future work we consider is deriving a theoretical bias analysis without the dependency on these assumptions.

# Acknowledgement

Work in this paper is supported by NSF ECCS 2412484.

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

# A  Assumptions and Real-World Graphs

In order to show that the assumptions made in Subsection 4.1 are typically valid for real-world graphs, here we present the exact values of the terms within Assumptions 2 and 3 for the datasets we use. Specifically, we make the following assumptions:

**A2:** $\frac{\mathbb{E}_{\tilde{\mathbf{A}}}[d_i^\omega]}{|\mathcal{S}_k|} \geq \frac{\mathbb{E}_{\tilde{\mathbf{A}}}[d_i^\mathcal{X}]}{N-|\mathcal{S}_k|}, \forall v_i \in \mathcal{S}_k, \forall k \in \{1, \cdots, K\}$,

**A3:** $\sum_{v_i, v_j \in \mathcal{V}} \tilde{A}_{ij} \gg \mathbb{E}_{\tilde{\mathbf{A}}}[d_i^\mathcal{X}], \forall v_i \in \mathcal{V}$.

First, for Assumption 2, we obtained the real values of the terms $l_k := \frac{\mathbb{E}_{\tilde{\mathbf{A}}, v_i \sim U_{\mathcal{S}_k}}[d_i^\omega]}{|\mathcal{S}_k|}$ and $r_k := \frac{\mathbb{E}_{\tilde{\mathbf{A}}, v_i \sim U_{\mathcal{S}_k}}[d_i^\mathcal{X}]}{N-|\mathcal{S}_k|}$, where we want $l_k \geq r_k \forall k \in \{1, \cdots, K\}$. For all different sensitive groups, these values are presented in Table 5 for all the used datasets.

Table 5: Validity of Assumption 2 for real-world graphs.

| Cora | $l_0$ | $l_1$ | $l_2$ | $l_3$ | $l_4$ | $l_5$ | $l_6$ | | | |
|---|---|---|---|---|---|---|---|---|---|---|
| | 0.0087 | 0.0174 | 0.0095 | 0.0035 | 0.0073 | 0.0094 | 0.0156 | | | |
| | $r_0$ | $r_1$ | $r_2$ | $r_3$ | $r_4$ | $r_5$ | $r_6$ | | | |
| | 0.0024 | 0.0023 | 0.0021 | 0.0021 | 0.0019 | 0.0019 | 0.018 | | | |
| Citeseer | $l_0$ | $l_1$ | $l_2$ | $l_3$ | $l_4$ | $l_5$ | | | | |
| | 0.0028 | 0.0026 | 0.0047 | 0.0026 | 0.0039 | 0.0034 | | | | |
| | $r_0$ | $r_1$ | $r_2$ | $r_3$ | $r_4$ | $r_5$ | | | | |
| | 0.0011 | 0.0012 | 0.0018 | 0.0011 | 0.0013 | 0.0009 | | | | |
| Amazon Photo | $l_0$ | $l_1$ | $l_2$ | $l_3$ | $l_4$ | $l_5$ | $l_6$ | $l_7$ | | |
| | 0.0874 | 0.0094 | 0.0428 | 0.0214 | 0.0260 | 0.0262 | 0.0202 | 0.0484 | | |
| | $r_0$ | $r_1$ | $r_2$ | $r_3$ | $r_4$ | $r_5$ | $r_6$ | $r_7$ | | |
| | 0.0055 | 0.0037 | 0.0045 | 0.0057 | 0.0054 | 0.0032 | 0.0088 | 0.0088 | | |
| Amazon Computer | $l_0$ | $l_1$ | $l_2$ | $l_3$ | $l_4$ | $l_5$ | $l_6$ | $l_7$ | $l_8$ | $l_9$ |
| | 0.0381 | 0.0078 | 0.0178 | 0.0281 | 0.0080 | 0.0891 | 0.0376 | 0.0219 | 0.0096 | 0.0696 |
| | $r_0$ | $r_1$ | $r_2$ | $r_3$ | $r_4$ | $r_5$ | $r_6$ | $r_7$ | $r_8$ | $r_9$ |
| | 0.0025 | 0.0033 | 0.0026 | 0.0034 | 0.0064 | 0.0024 | 0.0040 | 0.0017 | 0.0036 | 0.0024 |

For Assumption 3, we present the real values of the terms $\sum_{v_i, v_j \in \mathcal{V}} \tilde{A}_{ij}$ and $\mathbb{E}_{\tilde{\mathbf{A}}, v_i \sim U_\mathcal{V}}[d_i^\mathcal{X}]$, where we want $\sum_{v_i, v_j \in \mathcal{V}} \tilde{A}_{ij} \gg \mathbb{E}_{\tilde{\mathbf{A}}, v_i \sim U_\mathcal{V}}[d_i^\mathcal{X}]$. For all real-world datasets we use, these values are reported in Table 6.

Table 6: Validity of Assumption 3 for real-world graphs.

| | Cora | Citeseer | Amazon Photo | Amazon Computer |
|---|---|---|---|---|
| $\sum_{v_i, v_j \in \mathcal{V}} \tilde{A}_{ij}$ | 10556 | 9104 | 238162 | 491722 |
| $\mathbb{E}_{\tilde{\mathbf{A}}, v_i \sim U_\mathcal{V}}[d_i^\mathcal{X}]$ | 1.48 | 1.45 | 10.76 | 15.93 |

Overall, the results presented in both Tables 5 and 6 demonstrate that the assumptions made for the theoretical bias analysis in Section 4.1 are valid for the real-world graphs we are using. This supports that our analysis is applicable to most of the real-world data and settings.

# B  Proof of Theorem 1

Here, without loss of generality, we will focus on the $l$th GNN layer, where the input representations are represented by $\mathbf{H}^l$ and output representations are denoted $\mathbf{H}^{l+1}$. The considered disparity measure follows as:

$$\delta_k^{(l+1)} := \left\| \mathbb{E}_{\tilde{\mathbf{A}}, v_i \sim U_{\mathcal{S}_k}}[\mathbf{h}_i^{l+1} \mid s_i = k] - \mathbb{E}_{\tilde{\mathbf{A}}, v_i \sim U_{(\mathcal{V}-\mathcal{S}_k)}}[\mathbf{h}_i^{l+1} \mid s_i \neq k] \right\|_2. \tag{8}$$

Let's re-write the disparity measure $\delta_k^{(l+1)}$ by using definitions $\mathbf{c}_i^{l+1} := \mathbf{W}^l \mathbf{h}_i^l$, and $\mathbb{E}_{\tilde{\mathbf{A}}}[\mathbf{h}_i^{l+1}] = \sigma(\sum_{v_j \in \mathcal{V}} \tilde{A}_{ij} \mathbf{c}_j^{l+1})$.

$$
\begin{aligned}
\delta_k^{(l+1)} &:= \left\| \mathbb{E}_{\tilde{\mathbf{A}}, v_i \sim U_{\mathcal{S}_k}}[\mathbf{h}_i^{l+1} \mid s_i = k] - \mathbb{E}_{\tilde{\mathbf{A}}, v_i \sim U_{(\mathcal{V}-\mathcal{S}_k)}}[\mathbf{h}_i^{l+1} \mid s_i \neq k] \right\|_2, \\
&= \left\| \frac{1}{|\mathcal{S}_k|} \sum_{v_i \in \mathcal{S}_k} \sigma\left( \sum_{v_j \in \mathcal{V}} \tilde{A}_{ij} \mathbf{c}_j \right) - \frac{1}{N - |\mathcal{S}_k|} \sum_{v_i \notin \mathcal{S}_k} \sigma\left( \sum_{v_j \in \mathcal{V}} \tilde{A}_{ij} \mathbf{c}_j \right) \right\|_2.
\end{aligned}
\tag{9}
$$

Using Lemma A.1. in (57), it can be shown that $\delta_k^{(l+1)}$ can be upper-bounded as:

$$
\begin{aligned}
\delta_k^{(l+1)} &= \left\| \frac{1}{|\mathcal{S}_k|} \sum_{v_i \in \mathcal{S}_k} \sigma\left( \sum_{v_j \in \mathcal{V}} \tilde{A}_{ij} \mathbf{c}_j \right) - \frac{1}{N - |\mathcal{S}_k|} \sum_{v_i \notin \mathcal{S}_k} \sigma\left( \sum_{v_j \in \mathcal{V}} \tilde{A}_{ij} \mathbf{c}_j \right) \right\|_2, \\
&\leq L\left( \left\| \frac{1}{|\mathcal{S}_k|} \sum_{v_i \in \mathcal{S}_k} \sum_{v_j \in \mathcal{V}} \tilde{A}_{ij} \mathbf{c}_j - \frac{1}{N - |\mathcal{S}_k|} \sum_{v_i \notin \mathcal{S}_k} \sum_{v_j \in \mathcal{V}} \tilde{A}_{ij} \mathbf{c}_j \right\|_2 + 2\sqrt{N}\Delta_z \right).
\end{aligned}
\tag{10}
$$

Then, we can divide the sums over all nodes into two: nodes in $\mathcal{S}_k$ and the remaining ones.

$$
\begin{aligned}
\delta_k^{(l+1)} &\leq L\left( \left\| \frac{1}{|\mathcal{S}_k|} \sum_{v_i \in \mathcal{S}_k} \sum_{v_j \in \mathcal{V}} \tilde{A}_{ij} \mathbf{c}_j - \frac{1}{N - |\mathcal{S}_k|} \sum_{v_i \notin \mathcal{S}_k} \sum_{v_j \in \mathcal{V}} \tilde{A}_{ij} \mathbf{c}_j \right\|_2 + 2\sqrt{N}\Delta_z \right), \\
&= L\left( \left\| \left( \frac{1}{|\mathcal{S}_k|} \sum_{v_i \in \mathcal{S}_k} \sum_{v_j \in \mathcal{S}_k} \tilde{A}_{ij} \mathbf{c}_j + \frac{1}{|\mathcal{S}_k|} \sum_{v_i \in \mathcal{S}_k} \sum_{v_j \notin \mathcal{S}_k} \tilde{A}_{ij} \mathbf{c}_j \right) \right. \right. \\
&\quad \left. \left. - \left( \frac{1}{N - |\mathcal{S}_k|} \sum_{v_i \notin \mathcal{S}_k} \sum_{v_j \in \mathcal{S}_k} \tilde{A}_{ij} \mathbf{c}_j + \frac{1}{N - |\mathcal{S}_k|} \sum_{v_i \notin \mathcal{S}_k} \sum_{v_j \notin \mathcal{S}_k} \tilde{A}_{ij} \mathbf{c}_j \right) \right\|_2 + 2\sqrt{N}\Delta_z \right) \\
&= L\left( \left\| \sum_{v_j \in \mathcal{S}_k} \left( \frac{1}{|\mathcal{S}_k|} \sum_{v_i \in \mathcal{S}_k} \tilde{A}_{ij} - \frac{1}{N - |\mathcal{S}_k|} \sum_{v_i \notin \mathcal{S}_k} \tilde{A}_{ij} \right) \mathbf{c}_j \right. \right. \\
&\quad \left. \left. + \sum_{v_j \notin \mathcal{S}_k} \left( \frac{1}{|\mathcal{S}_k|} \sum_{v_i \in \mathcal{S}_k} \tilde{A}_{ij} - \frac{1}{N - |\mathcal{S}_k|} \sum_{v_i \notin \mathcal{S}_k} \tilde{A}_{ij} \right) \mathbf{c}_j \right\|_2 + 2\sqrt{N}\Delta_z \right)
\end{aligned}
\tag{11}
$$

Then, by triangle inequality, it follows that:

$$
\begin{aligned}
\delta_k^{(l+1)} &\leq L\left( \left\| \sum_{v_j \in \mathcal{S}_k} \left( \frac{1}{|\mathcal{S}_k|} \sum_{v_i \in \mathcal{S}_k} \tilde{A}_{ij} - \frac{1}{N - |\mathcal{S}_k|} \sum_{v_i \notin \mathcal{S}_k} \tilde{A}_{ij} \right) \mathbf{c}_j \right\|_2 \right. \\
&\quad \left. + \left\| \sum_{v_j \notin \mathcal{S}_k} \left( \frac{1}{|\mathcal{S}_k|} \sum_{v_i \in \mathcal{S}_k} \tilde{A}_{ij} - \frac{1}{N - |\mathcal{S}_k|} \sum_{v_i \notin \mathcal{S}_k} \tilde{A}_{ij} \right) \mathbf{c}_j \right\|_2 + 2\sqrt{N}\Delta_z \right).
\end{aligned}
\tag{12}
$$

Assumption 2 in Subsection 4.1 ensures that $\left( \frac{1}{|\mathcal{S}_k|} \sum_{v_i \in \mathcal{S}_k} \tilde{A}_{ij} - \frac{1}{N - |\mathcal{S}_k|} \sum_{v_i \notin \mathcal{S}_k} \tilde{A}_{ij} \right) \geq 0, \forall v_j \in \mathcal{S}_k$. Furthermore, third assumption presented in Subsection 4.1 guarantees that $\left( \frac{1}{|\mathcal{S}_k|} \sum_{v_i \in \mathcal{S}_k} \tilde{A}_{ij} - \frac{1}{N - |\mathcal{S}_k|} \sum_{v_i \notin \mathcal{S}_k} \tilde{A}_{ij} \right) \leq 0, \forall v_j \notin \mathcal{S}_k$. Utilizing Assumption 1 in Subsection 4.1, $\|\mathbf{c}_i\|_\infty \leq \delta, \forall v_i \in \mathcal{V}$, the following upper bound can be derived.

$$
\begin{aligned}
\delta_k^{l+1} &\leq L\left( \left\| \sum_{v_j \in \mathcal{S}_k} \left( \frac{1}{|\mathcal{S}_k|} \sum_{v_i \in \mathcal{S}_k} \tilde{A}_{ij} - \frac{1}{N - |\mathcal{S}_k|} \sum_{v_i \notin \mathcal{S}_k} \tilde{A}_{ij} \right) \boldsymbol{\delta}_{F^{(l+1)}} \right\|_2 \right. \\
&\quad \left. + \left\| \sum_{v_j \notin \mathcal{S}_k} \left( \frac{1}{|\mathcal{S}_k|} \sum_{v_i \in \mathcal{S}_k} \tilde{A}_{ij} - \frac{1}{N - |\mathcal{S}_k|} \sum_{v_i \notin \mathcal{S}_k} \tilde{A}_{ij} \right) \boldsymbol{\delta}_{F^{(l+1)}} \right\|_2 + 2\sqrt{N}\Delta_z \right),
\end{aligned}
\tag{13}
$$

where $\boldsymbol{\delta}_{F^{(l+1)}}$ stands for an $F^{(l+1)}$ dimensional vector with all elements being equal to $\delta$. Then, by utilizing the definitions $p_k^\chi := \sum_{v_i \in \mathcal{S}_k, v_j \notin \mathcal{S}_k} \tilde{A}_{i,j}, p_k^\omega := \sum_{v_i \in \mathcal{S}_k, v_j \in \mathcal{S}_k} \tilde{A}_{i,j}$, the upper bound in (13) can be rewritten as:

$$
\begin{aligned}
\delta_k^{(l+1)} \leq L\Bigg( & \left\| \sum_{v_j \in \mathcal{S}_k} \left( \frac{1}{|\mathcal{S}_k|} \sum_{v_i \in \mathcal{S}_k} \tilde{A}_{ij} - \frac{1}{N - |\mathcal{S}_k|} \sum_{v_i \notin \mathcal{S}_k} \tilde{A}_{ij} \right) \boldsymbol{\delta}_{F^{(l+1)}} \right\|_2 \\
& + \left\| \sum_{v_j \notin \mathcal{S}_k} \left( \frac{1}{|\mathcal{S}_k|} \sum_{v_i \in \mathcal{S}_k} \tilde{A}_{ij} - \frac{1}{N - |\mathcal{S}_k|} \sum_{v_i \notin \mathcal{S}_k} \tilde{A}_{ij} \right) \boldsymbol{\delta}_{F^{(l+1)}} \right\|_2 + 2\sqrt{N}\Delta_z \Bigg), \\
= L\Bigg( & \left\| \left( \frac{p_k^\omega}{|\mathcal{S}_k|} - \frac{p_k^\chi}{N - |\mathcal{S}_k|} \right) \boldsymbol{\delta}_{F^{(l+1)}} \right\|_2 + \left\| \left( \frac{\sum_{v_i,v_j \in \mathcal{V}} \tilde{A}_{ij} - p_k^\omega - 2p_k^\chi}{N - |\mathcal{S}_k|} - \frac{p_k^\chi}{|\mathcal{S}_k|} \right) \boldsymbol{\delta}_{F^{(l+1)}} \right\|_2 + 2\sqrt{N}\Delta_z \Bigg) \\
= L\Bigg( & \left| \frac{p_k^\omega}{|\mathcal{S}_k|} - \frac{p_k^\chi}{N - |\mathcal{S}_k|} \right| \|\boldsymbol{\delta}_{F^{(l+1)}}\|_2 + \left| \frac{\sum_{v_i,v_j \in \mathcal{V}} \tilde{A}_{ij} - p_k^\omega - 2p_k^\chi}{N - |\mathcal{S}_k|} - \frac{p_k^\chi}{|\mathcal{S}_k|} \right| \|\boldsymbol{\delta}_{F^{(l+1)}}\|_2 + 2\sqrt{N}\Delta_z \Bigg).
\end{aligned}
\tag{14}
$$

The final result follows from the inequality, $\|\boldsymbol{\delta}_{F^{(l+1)}}\|_2 \leq \delta\sqrt{F^{(l+1)}}$:

$$
\begin{aligned}
\delta_k^{(l+1)} := & \left\| \mathbb{E}_{\tilde{\mathbf{A}}, v_i \sim U_{\mathcal{S}_k}}[\mathbf{h}_i^{l+1} \mid s_i = k] - \mathbb{E}_{\tilde{\mathbf{A}}, v_i \sim U_{(\mathcal{V}-\mathcal{S}_k)}}[\mathbf{h}_i^{l+1} \mid s_i \neq k] \right\|_2, \\
\leq & L\left( \delta\sqrt{F^{(l+1)}} \left( \left| \frac{p_k^\omega}{|\mathcal{S}_k|} - \frac{p_k^\chi}{N - |\mathcal{S}_k|} \right| + \left| \frac{\sum_{v_i,v_j \in \mathcal{V}} \tilde{A}_{ij} - p_k^\omega - 2p_k^\chi}{N - |\mathcal{S}_k|} - \frac{p_k^\chi}{|\mathcal{S}_k|} \right| \right) + 2\sqrt{N}\Delta_z \right),
\end{aligned}
\tag{15}
$$

which concludes the proof.

## C  Proof of Proposition 1

Statistical parity for link prediction is defined as (19):

$$
\Delta_{\mathrm{SP}} := |\mathbb{E}_{(v_i,v_j) \sim U_\mathcal{V} \times U_\mathcal{V}}[g(v_i,v_j) \mid s_i = s_j] - \mathbb{E}_{(v_i,v_j) \sim U_\mathcal{V} \times U_\mathcal{V}}[g(v_i,v_j) \mid s_i \neq s_j]|.
\tag{16}
$$

By considering each sensitive group explicitly, statistical parity can also be written as:

$$
\Delta_{\mathrm{SP}} := \left| \sum_{k=1}^K \frac{|\mathcal{S}_k|}{N} \left( \mathbb{E}_{\tilde{\mathbf{A}}}[g(v_i,v_j) \mid s_i = s_j, s_i = k] - \mathbb{E}_{\tilde{\mathbf{A}}}[g(v_i,v_j) \mid s_i \neq s_j, s_i = k] \right) \right|.
\tag{17}
$$

Define $\mathbb{E}_{\tilde{\mathbf{A}}, v_i \sim U_{\mathcal{S}_k}}[\mathbf{h}_i \mid s_i = k] := \mathbf{p}_k$ and $\mathbb{E}_{\tilde{\mathbf{A}}, v_i \sim U_{(\mathcal{V}-\mathcal{S}_k)}}[\mathbf{h}_i \mid s_i \neq k] := \mathbf{q}_k$. We further assume that $\|\mathbf{h}_i\|_2 \leq q, \forall v_i \in \mathcal{V}$ and it holds that $\|\mathbb{E}_{\tilde{\mathbf{A}}, v_i \sim U_{\mathcal{S}_k}}[\mathbf{h}_i \mid s_i = k] - \mathbb{E}_{\tilde{\mathbf{A}}, v_j \sim U_{(\mathcal{V}-\mathcal{S}_k)}}[\mathbf{h}_j \mid s_j \neq k]\|_2 \leq \delta_k^{max}$. Using the definitions for $\mathbf{p}_k$ and $\mathbf{q}_k$ and link prediction model $g(v_i,v_j) := \mathbf{h}_i^\top \Sigma \mathbf{h}_j$, $\Delta_{\mathrm{SP}}$ can be reformulated as:

$$
\begin{aligned}
\Delta_{\mathrm{SP}} := & \left| \sum_{k=1}^K \frac{|\mathcal{S}_k|}{N} \left( \mathbf{p}_k^\top \Sigma \mathbf{p}_k - \mathbf{p}_k^\top \Sigma \mathbf{q}_k \right) \right|, \\
= & \left| \sum_{k=1}^K \frac{|\mathcal{S}_k|}{N} \left( \mathbf{p}_k^\top \Sigma (\mathbf{p}_k - \mathbf{q}_k) \right) \right|.
\end{aligned}
\tag{18}
$$

By triangle inequality, it follows that

$$
\Delta_{\mathrm{SP}} \leq \sum_{k=1}^K \frac{|\mathcal{S}_k|}{N} \left| \left( \mathbf{p}_k^\top \Sigma (\mathbf{p}_k - \mathbf{q}_k) \right) \right|.
\tag{19}
$$

Finally, by using Cauchy-Schwarz inequality and the assumption $\|\mathbf{h}_i\|_2 \leq q, \forall v_i \in \mathcal{V}$, we can conclude that

$$\Delta_{\text{SP}} \leq \sum_{k=1}^{K} \frac{|\mathcal{S}_k|}{N} q \|\Sigma\|_2 \left\| (\mathbf{p}_k - \mathbf{q}_k) \right\|_2,$$

$$\Delta_{\text{SP}} \leq \sum_{k=1}^{K} \frac{|\mathcal{S}_k|}{N} q \|\Sigma\|_2 \delta_k^{max}, \tag{20}$$

where the final inequality follows from the assumption that $\|\mathbb{E}_{\tilde{\mathbf{A}}, v_i \sim U_{\mathcal{S}_k}}[\mathbf{h}_i \mid s_i = k] - \mathbb{E}_{\tilde{\mathbf{A}}, v_j \sim U_{(\mathcal{V} - \mathcal{S}_k)}}[\mathbf{h}_j \mid s_j \neq k]\|_2 \leq \delta_k^{max}$.

## D  Diffusion Model

**Diffusion Models for Graph Generation.**  Briefly, diffusion models are composed of two main elements: a noise model $q$, and a denoising neural network $\phi_\theta$. The noise model $q$ progressively corrupts data to create a sequence of increasingly noisy data points. Inspired by the success of Gaussian noise for diffusion-based image generation (58), the earlier diffusion-based graph generation models employ Gaussian noise to create noisy graph data (45; 46). However, such a noise model cannot properly capture the structural properties of discrete graph connections. Motivated by this, a discrete noise model is introduced in (51). The discrete noise model for graph structure is typically applied in the form of edge deletion and additions (16; 47). After creating noisy data, a denoising network $\phi_\theta$ is trained to invert this process by predicting the original graph structure $\mathbf{A}$ from the noisy samples. While different neural network structures are examined as candidates for the denoising network, message-passing neural networks are shown to be a scalable solution for the creation of medium- to large-scale graphs (47).

**Forward diffusion process:**  introduced in (51), we employ a Markov process herein to add noise to the input graph structure in the form of edge additions or deletions. These edge modifications can be executed by modeling the existence/non-existence of an edge as the edge class labels, where we have 2 classes, and applying a transition matrix that switches the labels with a certain probability. Then, given $\mathbf{A}^0 \in \mathbb{R}^{N \times N \times 2}$ denotes the one-hot representations for the edge labels, the noise model can be described by the transition matrices $\mathbf{Q}_A^t \in \mathbb{R}^{2 \times 2}$ for $t = 1, \cdots, T$, where $q(\mathbf{A}^t \mid \mathbf{A}^{t-1}) = \mathbf{A}^{t-1} \mathbf{Q}_A^t$, $q(\mathbf{A}^t \mid \mathbf{A}^0) = \mathbf{A}^0 \bar{\mathbf{Q}}_A^t$, and $\bar{\mathbf{Q}}_A^t = \mathbf{Q}_A^1 \mathbf{Q}_A^2 \cdots \mathbf{Q}_A^t$. For a uniform transition model, (59) proves that the empirical data distribution (the probability for the existence of an edge) is the optimal prior distribution. Following this finding, we specifically employ the transition matrix:

$$\mathbf{Q}_A^t = \alpha^t \mathbf{I} + (1 - \alpha^t) \mathbf{1} \mathbf{m}_E^\top, \tag{21}$$

where $\mathbf{I} \in \mathbb{R}^{2 \times 2}$ is the identity matrix, $\mathbf{1} \in \mathbb{R}^2$ is a vector of ones, and $\mathbf{m}_E \in \mathbb{R}^2$ describes the distribution of edge labels in the original graph. For the assignment of $\alpha^t$, cosine schedule is utilized, where $\bar{\alpha}^t := \cos(0.5\pi(t/T + s)/(1 + s))^2$ with a small s value for $\bar{\alpha}^t = \Pi_{\tau=1}^t \alpha^\tau$. Note that for categorical nodal features, the forward diffusion process follows the same procedure.

**Sampling:**  Using the trained MPNN, synthetic graphs can be sampled iteratively. During sampling, we first sample the sensitive attributes of the nodes, $\tilde{\mathbf{S}}$, based on its original distribution in the real graph. As the next step, we need to estimate the reverse diffusion iterations $p_\theta \left( \mathcal{G}^{t-1} = (\mathbf{A}^{t-1}, \mathbf{X}^{t-1}) \mid \mathcal{G}^t = (\mathbf{A}^t, \mathbf{X}^t) \right)$, which is modeled as a product over nodes and edges (59):

$$p_\theta \left( \mathcal{G}^{t-1} \mid \mathcal{G}^t, \tilde{\mathbf{S}}, t \right) = \prod_{i=1}^{N} \prod_{f=1}^{F} p_\theta \left( \mathbf{X}_{if}^{t-1} \mid \mathcal{G}^t, \tilde{\mathbf{S}}, t \right)$$

$$\prod_{1 \leq i < j \leq N} p_\theta \left( \mathbf{A}_{ij}^{t-1} \mid \mathcal{G}^t, \tilde{\mathbf{S}}, t \right). \tag{22}$$

To compute each independent term, we marginalize over the predictions of the MPNN:

$$p_\theta \left( \mathbf{A}_{ij}^{t-1} \mid \mathcal{G}^t, \tilde{\mathbf{S}}, t \right) = \sum_{e \in \mathcal{E}^c} p_\theta \left( \mathbf{A}_{ij}^{t-1} \mid \mathbf{A}_{ij} = e, \mathcal{G}^t \right) p_\theta \left( \mathbf{A}_{ij} = e \mid \mathcal{G}^t, \tilde{\mathbf{S}}, t \right)$$

$$\approx \sum_{e \in \mathcal{E}^c} p_\theta \left( \mathbf{A}_{ij}^{t-1} \mid \mathbf{A}_{ij} = e, \mathcal{G}^t \right) \tilde{\mathbf{A}}_{ije}, \tag{23}$$

where $\mathcal{E}^c$ denotes all possible labels for edges, which are $\{0, 1\}$ for an unweighted graph. Then, we can use our Markovian noise to model $p_\theta \left( \mathbf{A}_{ij}^{t-1} \mid \mathbf{A}_{ij} = e, \mathcal{G}^t \right)$:

$$p_\theta \left( \mathbf{A}_{ij}^{t-1} \mid \mathbf{A}_{ij} = e, \mathcal{G}^t \right) = \begin{cases} q \left( \mathbf{A}_{ij}^{t-1} \mid \mathbf{A}_{ij} = e, \mathbf{A}_{ij}^t \right) & \text{if } q \left( \mathbf{A}_{ij}^t \mid \mathbf{A}_{ij} = e \right) > 0 \\ 0 & \text{otherwise.} \end{cases}$$

Note that posterior $q \left( \mathbf{A}_{ij}^{t-1} \mid \mathbf{A}_{ij} = e, \mathbf{A}_{ij}^t \right)$ can be computed in closed-form using Bayes rule. Leveraging this model, $G^{t-1}$ can be sampled, which becomes the input of the MPNN at the next time step.

## E   Additional Details on Datasets and Datasets Statistics

For citation networks (Cora and Citeseer), the one-hot encoding representations of the words in the article descriptions constitute the binary nodal attributes. In these networks, similar to the setups in (19; 37), the category of the articles is employed as the sensitive attribute. Furthermore, for product co-purchase networks, nodal attributes are again the one-hot encodings of the words in the product reviews and the product categories are utilized as the sensitive attributes. For the evaluation of node classification, we employ German (56) and Pokec-n (9) networks. Specifically, the German credit graph has 1,000 nodes representing the clients in a German bank, where the links are created based on the similarity of credit accounts. For this graph, the node labels classify clients into good vs. bad credit risks, where the clients' gender are employed as the sensitive attribute (28). In addition, Pokec-n is sampled from an anonymized version of the Pokec network of 2012 (a social network from Slovakia), where nodes correspond to users who live in two major regions, and the region information is utilized as the sensitive attribute (9). The working field of the users is binarized and utilized as the labels to be predicted in node classification.

Table 7: Dataset statistics.

| Dataset | $|\mathcal{V}|$ | $|\mathcal{E}|$ | $F$ | $K$ |
|---|---|---|---|---|
| Cora | 2708 | 10556 | 1433 | 7 |
| Citeseer | 3327 | 9228 | 3703 | 6 |
| Amazon Photo | 7650 | 238163 | 745 | 8 |
| Amazon Computer | 13752 | 491722 | 767 | 10 |
| Credit | 1000 | 22242 | 27 | 2 |
| Pokec-n | 6185 | 21844 | 59 | 2 |

Statistical information for the utilized datasets are presented in Table 7, where $F$ is the total number of nodal features and $K$ represents the number of sensitive groups.

## F   Evaluation with Statistics

We evaluate the created synthetic graphs by FairWire via link prediction in Subsection 6.2. In order to provide a more traditional evaluation scheme, here we also report the 1-Wasserstein distance between the node degree distribution and clustering coefficient distribution of original graph and the synthetic ones. Table 8 presents the corresponding distance measures, where lower values for all metrics signify better performance. In Table 8, ER and SBM stand for traditional baselines Erdos–Rényi model (60) anf SBM stochastic block model (61), respectively. Furthermore, as deep learning based baselines, Feature-based MF represents feature-based matrix factorization (62), GAE and VGAE stand for graph autoencoder and variational graph autoencoder (63), respectively. Finally, GraphMaker in Table 8 corresponds to a diffusion-based graph generation baseline (47). Note that all these baselines are fairness-agnostic. Overall, results in Table 8 signify that FairWire can create synthetic graphs that follow a similar distribution to the original data while also improving the fairness metrics.

## G   Implementation Details

**Link Prediction Model.** For link prediction, we train a one-layer graph convolutional network (GCN), where the inner product between the output node representations signifies the corresponding edge probability. For training, $80\%$ of the edges are used, where the remaining edges are split equally into two for the validation and test sets. For link prediction experiments, results are obtained for five

|  | Cora | | Citeseer | | Amazon Photo | |
|---|---|---|---|---|---|---|
|  | Degree ↓ | Cluster↓ | Degree↓ | Cluster↓ | Degree↓ | Cluster↓ |
| ER | 1.0 | $2.4e^1$ | $8.5e^{-1}$ | $1.4e^1$ | $1.9e^1$ | $4.0e^1$ |
| SBM | $9.6e^{-1}$ | $2.3e^1$ | $8.0e^{-1}$ | $1.4e^1$ | $1.5e^1$ | $3.8e^1$ |
| Feature-based MF | $1.3e^3$ | $2.4e^1$ | $1.7e^3$ | $1.4e^1$ | $3.8e^3$ | $4.0e^1$ |
| GAE | $1.3e^3$ | $2.4e^1$ | $1.7e^3$ | $1.4e^1$ | $3.8e^3$ | $4.0e^1$ |
| VGAE | $1.4e^3$ | $2.4e^1$ | $1.7e^3$ | $1.4e^1$ | $3.8e^3$ | $4.0e^1$ |
| GraphMaker | 2.8 | $2.3e^1$ | $5.6e^{-1}$ | $1.4e^1$ | $1.1e^2$ | 1.4 |
| FairWire | 1.9 | $2.4e^1$ | $8.7e^{-1}$ | $1.4e^1$ | $8.2e^1$ | 1.5 |

Table 8: Distances of statistical measures between the real graph and synthetic ones.

random data splits, and their average along with the standard deviations are reported. The weights of the GNN model for link prediction are initialized utilizing Glorot initialization (64), where it is trained for 1000 epochs by employing Adam optimizer (65). The learning rate, the dimension of hidden representations, and the dropout rate are selected via grid search for the proposed scheme and all baselines, where the value leading to the best validation set performance is selected. For learning rate the, the dimension of hidden representations, and the dropout rate, the corresponding hyperparameter spaces are $\{1e-1, 1e-2, 3e-3, 1e-3\}$, $\{32, 128, 512\}$, and $\{0.0, 0.1, 0.2\}$, respectively.

**Diffusion Model.** Diffusion models are trained for 10000 epochs by employing Adam optimizer (65), where the number of diffusion steps is 3. In the MPNN described in Subsection 5.2, hidden representation size for time step $t$ is 32 for Cora and Citeseer and 16 for the Amazon Photo, German credit, and Pokec-n networks. In addition, hidden representation sizes for the nodal attributes and sensitive attributes are 512 and 64, respectively, for all datasets. The MPNN consists of two layers.

**Node Classification Model.** For node classification, we employ one-layer and two-layers APPNP (66) networks for German credit and Pokec-n graphs, respectively. For training, 50% of the nodes are used, where the remaining nodes are split equally into two for the validation and test sets. The weights of the GNN model for node classification are initialized utilizing Glorot initialization (64), where it is trained for 1000 epochs by employing Adam optimizer (65). The learning rate, the dimension of hidden representations, and the dropout rate are selected via grid search for the proposed scheme and all baselines, where the value leading to the best validation set performance is selected. For learning rate the, the dimension of hidden representations, and the dropout rate, the corresponding hyperparameter spaces are $\{3e-2, 1e-2, 3e-3\}$, $\{32, 128, 512\}$, and $\{0.0, 0.1\}$, respectively.

**Hyperparameter Selection.** For the link prediction task, we select the multiplier of $\mathcal{L}_{\text{FairWire}}$ among the values $\{0.01, 0.05, 0.1, 0.5\}$ via grid search (the multiplier of the cross-entropy loss is 1). The results for $\mathcal{L}_{\text{FairWire}}$ in Table 2 are obtained for the $\lambda$ values $0.05, 0.1, 0.01/0.05, 0.1$ on Cora, Citeseer, Amazon Photo, and Amazon Computer, respectively. For adversarial regularization (9), the multiplier of the regularizer is selected via a grid search among the values $\{0.1, 1, 10\}$ (the multiplier of link prediction loss is again 1). The multiplers of the adversarial regularization for the results in Table 2 are $\{10, 1, 1, 1\}$ on Cora, Citeseer, Amazon Photo, and Amazon Computer, respectively. Furthermore, the hyperparameter $\delta$ in FairDrop algorithm is tuned among the values $\{0.16, 0.25\}$ (0.16 is the default value in their codes), where 0.16 led to the best fairness/utility trade-off on each dataset. For FairAdj (19), we use the hyperparameter values suggested by (19) directly for the citation networks.

For the generative models, we select the multiplier of $\mathcal{L}_{\text{FairWire}}$ ($\lambda$ in (7)) among the values $\{0.05, 0.1, 1.0, 10.0\}$ via grid search. The results for FairWire in Table 3 are reported for the $\lambda$ values $10.0, 0.1$ on Cora, and Citeseer, respectively ($\lambda = 0.05$ for Amazon photo in Table 11). For the results in Table 4, $\lambda$ equals to 10 and 1 for German credit and Pokec-n, respectively. For adversarial regularization (9), the multiplier of the regularizer (again in the training loss of the MPNN) is selected via a grid search among the values $\{0.001, 0.01, 0.1\}$. The multipliers of the adversarial regularization for the results in Table 3 are $\{0.01, 0.01, 0.01\}$ on Cora, Citeseer, and Amazon Photo, respectively. Furthermore, the hyperparameter $\eta$ in FairAdj (19) algorithm is tuned among the values $\{0.001, 0.005, 0.01\}$, where 0.001 led to the best fairness/utility trade-off on each dataset.

**Baselines.** Fairness-aware baselines in the experiments include adversarial regularization (9), FairDrop (37), FairAdj (19), and FairGen (49). Adversarial regularization refers to the bias mitigation technique where an adversary is trained to predict the sensitive attributes. In link prediction, the

adversary is trained to predict the sensitive attributes of the nodes that are incident to the edges whose labels are of interest. Furthermore, FairDrop (37) is an edge dropout strategy where the dropout probabilities vary for intra- or inter-edges so as to create a more balanced graph topology. FairAdj (19) optimizes a fair adjacency with certain structural constraints for link prediction in an iterative manner considering both fairness and utility. Finally, FairGen (35) focuses on the disparities in generation quality (distances between different graph statistics) for different sensitive groups and employs fairness-aware regularizers during graph generation via a transformer-based model.

For graph generation experiments, GraphMaker (47) is utilized to create synthetic graphs in a fairness-agnostic way. While an existing work that considers fair link prediction for synthetic graphs is not available to the best of our knowledge, we employ adversarial regularization (9) as an in-processing bias mitigation strategy during the training of the MPNN described in Subsection 5.2. Furthermore, we use FairAdj (19) as a post-processing bias mitigation strategy on the synthetic graphs generated via GraphMaker, and the processed synthetic graphs are then evaluated for the link prediction and node classification tasks.

# H   Sensitivity Analysis

In order to examine the impact of hyperparameter selection on fairness improvements, the sensitivity analyses for the proposed tools are executed with respect to the hyperparameter $\lambda$. The results are obtained for changing $\lambda$ values for both the link prediction (see (4)) and graph generation (see (7)) experiments and reported in Tables 9, 10. Overall, the results signify that both $\mathcal{L}_{\text{FairWire}}$ in the link prediction and FairWire, lead to better fairness measures compared to the natural baselines within a wide range of hyperparameter selection.

Table 9: Sensitivity Analyses for Different Tasks

| | Cora | | | Citeseer | | |
|---|---|---|---|---|---|---|
| **Link Prediction** | AUC (%) | $\Delta_{SP}$ (%) | $\Delta_{EO}$ (%) | AUC (%) | $\Delta_{SP}$ (%) | $\Delta_{EO}$ (%) |
| GNN | $\mathbf{94.43} \pm 0.74$ | $27.01 \pm 1.38$ | $9.11 \pm 1.43$ | $\mathbf{96.16} \pm 0.28$ | $27.40 \pm 1.24$ | $7.37 \pm 1.33$ |
| $\lambda = 0.01$ | $93.85 \pm 1.06$ | $16.17 \pm 4.50$ | $5.58 \pm 1.18$ | $95.55 \pm 0.66$ | $17.38 \pm 7.80$ | $6.55 \pm 2.33$ |
| $\lambda = 0.05$ | $92.18 \pm 1.03$ | $4.76 \pm 0.24$ | $2.05 \pm 0.37$ | $\mathbf{96.18} \pm 0.46$ | $12.43 \pm 0.57$ | $5.44 \pm 0.86$ |
| $\lambda = 0.1$ | $91.98 \pm 1.05$ | $\mathbf{4.54} \pm 0.24$ | $\mathbf{1.95} \pm 0.36$ | $96.00 \pm 0.23$ | $\mathbf{8.62} \pm 0.80$ | $\mathbf{1.29} \pm 0.68$ |

| | Amazon Photo | | | Amazon Computer | | |
|---|---|---|---|---|---|---|
| **Link Prediction** | AUC (%) | $\Delta_{SP}$ (%) | $\Delta_{EO}$ (%) | AUC (%) | $\Delta_{SP}$ (%) | $\Delta_{EO}$ (%) |
| GNN | $97.01 \pm 0.26$ | $32.65 \pm 0.95$ | $8.01 \pm 0.52$ | $\mathbf{96.13} \pm 0.06$ | $23.70 \pm 0.79$ | $5.51 \pm 0.79$ |
| $\lambda = 0.01$ | $\mathbf{97.25} \pm 0.11$ | $27.75 \pm 0.52$ | $7.11 \pm 0.41$ | $96.13 \pm 0.08$ | $23.58 \pm 0.63$ | $5.46 \pm 0.56$ |
| $\lambda = 0.05$ | $94.85 \pm 0.32$ | $24.61 \pm 0.96$ | $6.24 \pm 1.22$ | $96.17 \pm 0.13$ | $22.74 \pm 0.39$ | $5.50 \pm 0.51$ |
| $\lambda = 0.1$ | $88.43 \pm 5.12$ | $\mathbf{17.78} \pm 7.27$ | $\mathbf{1.48} \pm 1.00$ | $92.66 \pm 0.21$ | $\mathbf{14.45} \pm 0.32$ | $\mathbf{1.28} \pm 0.45$ |

| | Citeseer | | | Amazon Photo | | |
|---|---|---|---|---|---|---|
| **Graph Generation** | AUC (%) | $\Delta_{SP}$ (%) | $\Delta_{EO}$ (%) | AUC (%) | $\Delta_{SP}$ (%) | $\Delta_{EO}$ (%) |
| $\tilde{\mathcal{G}}$ | $\mathbf{92.19} \pm 1.06$ | $37.56 \pm 1.29$ | $13.52 \pm 0.92$ | $\mathbf{94.45} \pm 0.21$ | $33.49 \pm 0.28$ | $10.01 \pm 0.56$ |
| $\lambda = 0.01$ | $92.03 \pm 0.80$ | $36.43 \pm 1.44$ | $13.11 \pm 0.71$ | $93.41 \pm 0.34$ | $27.18 \pm 5.45$ | $5.45 \pm 0.80$ |
| $\lambda = 0.05$ | $91.89 \pm 1.37$ | $29.40 \pm 4.34$ | $9.56 \pm 1.29$ | $93.88 \pm 0.40$ | $25.27 \pm 0.76$ | $3.13 \pm 0.30$ |
| $\lambda = 0.1$ | $91.27 \pm 2.78$ | $\mathbf{18.35} \pm 6.91$ | $\mathbf{7.80} \pm 2.66$ | $88.43 \pm 5.12$ | $\mathbf{17.78} \pm 7.27$ | $\mathbf{1.48} \pm 1.00$ |

Table 10: Sensitivity Analysis for Graph Generation on Cora

| | Cora | | |
|---|---|---|---|
| | AUC (%) | $\Delta_{SP}$ (%) | $\Delta_{EO}$ (%) |
| $\tilde{\mathcal{G}}$ | $87.29 \pm 1.09$ | $35.72 \pm 1.74$ | $13.27 \pm 0.81$ |
| $\lambda = 1.0$ | $\mathbf{88.76} \pm 1.23$ | $27.86 \pm 2.84$ | $10.76 \pm 1.50$ |
| $\lambda = 10.0$ | $86.49 \pm 2.79$ | $12.91 \pm 6.35$ | $4.31 \pm 3.59$ |
| $\lambda = 50.0$ | $82.91 \pm 8.06$ | $\mathbf{9.86} \pm 6.54$ | $\mathbf{3.47} \pm 2.58$ |

# I  Additional Graph Generation Results

Due to limited space, we present the comparative results for graph generation on Amazon Photo in Table 11 for link prediction.

Table 11: Graph generation results on Amazon-Photo

|  | Amazon Photo | | |
|---|---|---|---|
|  | AUC (%) | $\Delta_{SP}$ (%) | $\Delta_{EO}$ (%) |
| $\mathcal{G}$ | 96.91 | 32.58 | 8.24 |
| $\tilde{\mathcal{G}}$ | **94.45** $\pm$ 0.21 | 33.49 $\pm$ 0.28 | 10.01 $\pm$ 0.56 |
| Adversarial | 94.24 $\pm$ 1.20 | 29.17 $\pm$ 2.83 | 7.06 $\pm$ 2.63 |
| FairWire | 93.88 $\pm$ 0.40 | **25.27** $\pm$ 0.76 | **3.13** $\pm$ 0.30 |

Similar to the link prediction experiments (Table 2), the results of FairAdj for the Amazon Photo network could not be obtained due to computational limitations. Overall, the results reported in Table 11 conclude again that FairWire can provide the best utility/fairness trade-off compared to other fairness-aware baselines.

# J  Computational Resources

Experiments are carried over on 4 NVIDIA RTX A4000 GPUs.

