# OpenReview forum: "FairWire: Fair Graph Generation"
_NeurIPS.cc/2024/Conference — NeurIPS 2024 poster_

### Official Review · Reviewer_bPaX · 2024-07-10

**Soundness:** 3
**Presentation:** 3
**Contribution:** 3
**Rating:** 7
**Confidence:** 4

**Summary:**

In this submission a graph diffusion model with fairness correction is introduced for graph generation. The model is also applied for link prediction. The authors introduce a graph regularizer for fairness based on a theoretical bound for subgroup distance and representation distance. This analysis is newly introduced by the authors. The authors apply the regularizer to link prediction, examining the accuracy-fairness tradeoff. The authors then test fair graph generation by examining link prediction and node classification tasks, but replace the training graphs with fairness-corrected generated graphs. Empirically it is shown that this greatly improves fairness while marginally reducing accuracy.

**Strengths:**

This paper has key straightforward strengths that contribute to my "weak accept" rating. First, the method seems to perform well, within the scope of the intended contributions. The paper is generally well-written and the presentation is clear. Furthermore the theoretical results are satisfactory. They appear correct and they properly motivate the method.

**Weaknesses:**

There is some missing/glossing-over existing work. The authors claim that the theoretical analysis is novel, however ref [24] in the paper also includes a theoretical analysis with >2 binary categories. It would be good for the authors to discuss this work and compare with their own. Also, the authors should consider citing "Debiasing Graph Representations via Metadata-Orthogonal Training" (doi/10.1109/ASONAM49781.2020.9381348), which seems relevant to the problem (also includes a fairness correction over potentially >2 attributes).

There are a non-trivial amount of clarity and presentation issues, which I discuss in my list of questions.

**Questions:**

(1) In the second experiment type (S6.3), which link prediction model was trained on the generated graphs? And was the FairWire regularization also applied to the link prediction model during training?

(2) From Fig 1 we can see that FairWire effectively removes intra-edges from generated graphs. However, the structure of the generate graphs is not quite evident. Can the graphs be visualized in a more productive way? Maybe the adjacencies can be row-sorted into their communities.

(3) The authors derive two fairness criterion $\alpha_1$ and $\alpha_2$, though $\alpha_2$ is ignored at the mini-batch level for scalability. Can the authors give more intution about $\alpha_2$ and why it should ultimately not matter when applied in practice?

**Limitations:**

Yes

---

> ### Author Rebuttal · Authors · 2024-08-06
>
> We would like to sincerely thank the Reviewer for the raised points. We have addressed all points raised by the Reviewer, and presented our responses below.
>
> **Weakness:** We thank the Reviewer for this comment under Weaknesses.
>
> In [24], it is stated that:
>
> “For clarity and simplicity, we consider the case of a single binary sensitive attribute, with the theoretical intuitions naturally generalizing to the multiattribute and multi-class settings.”. Hence, in their theoretical analysis, [24] considers a single, binary sensitive attribute, while we take multi-valued sensitive attributes into account in our analysis.
>
> In addition, we want to clarify that our analysis is the first theoretical investigation for the relation between $\Delta _ {\mathrm{SP}}$ and the graph topology considering multi-valued sensitive attributes. The analysis in [24] concludes that if the weight of the adversarial regularizer is set to infinity, the equilibrium of their objective is achieved when there is zero mutual information between the sensitive attribute and the node embeddings. However, setting the weight of the adversarial regularizer to infinity is impractical, which limits the applicability of their analysis. Thus, there are significant differences in the theoretical findings of [24] and our work, where we specifically focus on structural bias and its analytic relation to a well-accepted bias metric $\Delta _ {\mathrm{SP}}$ for non-binary sensitive attributes, which was not explored before to the best of our knowledge.
>
> We thank the Reviewer for pointing out this related work (doi/10.1109/ASONAM49781.2020.9381348), which we will cite in the final submission.
>
> **Q1:** Thank you for this insightful question.
>
> The same link prediction model is used for link prediction (S6.2) and graph generation (S6.3) experiments, whose details are provided in Appendix G. Furthermore, we did not apply any extra fairness intervention on FairWire during link prediction model training (including FairWire regularization). FairWire only modifies the training for graph generation, as we wanted to evaluate the effectiveness of structural debiasing that is executed in a task-agnostic way.
>
> **Q2:** Thank you for this suggestion.
>
> Based on this comment, we created the visualizations for row-sorted adjacencies (based on sensitive attributes) of the real Cora network and its synthetic version created by FairWire. The created figures are presented in the attached PDF document to the General Rebuttal. The figures confirm our previous finding that, with the employment of FairWire, the links between nodes become less correlated with the sameness of sensitive attributes of the nodes. Specifically, for the real graph, most links are built within the same sensitive group (diagonal entries), while the link formation becomes more uniform across different sensitive groups after applying FairWire.
>
> In addition, in order to provide structural information, we present the 1-Wasserstein distance between the node degree distribution and clustering coefficient distribution of the original graph and the synthetic graphs created by FairWire in Appendix F.
>
> We hope this reply addresses your concerns.
>
> **Q3:** Thank you for raising this point. We would like to clarify that our submission does not conclude or claim that $\alpha _ {2}$ does not matter for fairness. In fact, the consideration of $\alpha _ {2}$ together with $\alpha _ {1}$ might further benefit the fairness. The corresponding solutions for $( p _ {k} ^ { \omega } ) ^ { * }$ and $(p _ {k} ^ {\chi}) ^ {*}$ for this case are presented in Subsection 4.2. However, as the reviewer mentioned, a regularizer that is designed based on both $\alpha _ {1}$ and $\alpha _ {2}$ cannot be employed in a mini-batch setting, which limits its use, especially for generative models. Since the main focus of FairWire is mitigating structural bias in graph generation, which can only be trained in a mini-batch setting for medium to large-scale graphs due to a complexity growing exponentially with the number of nodes, we focused on a design that can be used in a broader range of learning settings. Our experimental results confirm the effectiveness of the proposed fairness regularizer, which typically provides the best fairness/utility trade-off compared to state-of-the-art fairness-aware baselines.

---

> > ### Comment · Reviewer_bPaX · 2024-08-08
> > **Thanks**
> >
> > The authors have satisfactorily addressed my concerns. I have raised the presentation score and the overall score.

---

### Official Review · Reviewer_miG4 · 2024-07-12

**Soundness:** 3
**Presentation:** 3
**Contribution:** 3
**Rating:** 7
**Confidence:** 3

**Summary:**

This work considers the problem of fairness in learning over graphs. Namely, it adopts a criterion for fairness that quantifies the probability of a relationship existing between nodes whose sensitive attribute value matches (intra-edges) versus not (inter-edges). It first derives theoretical results that bound the discrepancy between the two, and uses these insights to design a "fairness regularizer" that is compatible with both link prediction and synthetic graph generation tasks. The authors perform an evaluation on these two tasks, outperforming existing baselines in fairness metrics while preserving the core utility metric (e.g., accuracy).

**Strengths:**

The paper considers a problem that has been studied previously but is original in the way it approaches it. The quality of the work is high and contains both theoretical insights and thorough experiments. The organisation of the paper is clear and the writing quality is also high. The topic is significant and relevant to the NeurIPS community. Reproducibility is also good as code is provided and experiments are described clearly.

**Weaknesses:**

The paper is very well executed. In my opinion, its main weakness is that the considered fairness definition is adopted without a convincing justification. It is rather well-known in the network science literature [1,2] that individuals tend to form connections with those that are similar to them. Therefore, I do not understand why this characteristic is framed as inherently "bad" and something that should be corrected when training a model for link prediction or graph generation. I think this deserves a proper justification and discussion. However, this may be a point that is broader than the paper itself.

[1] McPherson, M., Smith-Lovin, L., & Cook, J. M. (2001). Birds of a feather: Homophily in social networks. Annual review of sociology, 27(1), 415-444.

[2] Newman, M. E. (2003). Mixing patterns in networks. Physical review E, 67(2), 026126.

**Questions:**

Please address the potential weakness discussed above. Additional questions and comments:

C1. Could you discuss what happens if there is >1 sensitive attribute? Do your results already apply via some transformation, or what would be required to generalise to this case?

C2. Line 131-32: I think this claim needs to be scoped to e.g. *social* networks, as this is definitely not always the case (see e.g. ref [2] above).

C3. Line 153: maximization is over $k$ presumably?

C4. Line 167; Eq 3: $\mathbf{Se}_{k}$ not defined as far as I can tell.

C5. Experiments in 6.2: I think it should be discussed what "base GNN" is being used. Appendix G mentions it is a GCN -- but what is the justification for using a single layer?

C6. I think it's worth summarizing the choice and justification for the considered sensitive attributes in the main text (currently in Appendix E).

C7. Table 5 is overflowing the margins, consider wrapping in `\resizebox`

**Limitations:**

Limitations and potential negative social impacts are discussed, although the latter only superficially, and I think it deserves more discussion given the primary area of the paper is fairness.

---

> ### Author Rebuttal · Authors · 2024-08-06
>
> We would like to thank the Reviewer for the supportive and constructive remarks, as well as valuable comments. We have presented our responses to the Reviewer’s questions below.
>
> **Weakness:** Regarding the comment under Weaknesses, we agree with the reviewer that “individuals tend to form connections with those that are similar to them,” and we would like to clarify that such links between similar nodes in graphs are not necessarily bad, instead such connectedness generally provides additional information and is leveraged in graph ML. However, while admittedly entailing useful information for learning, the homophilic relations built based on sensitive attributes may amplify the bias in predictions [R1]. For example, it has been shown that ad recommenders display racial biases between users with similar preferences [R2], where the denser connectivity between the users from the same ethnicity in the corresponding social networks might amplify this issue. Our proposed framework aims to mitigate the structural bias leading to such discriminatory predictions in the decision systems requiring fairness considerations.
>
> **C1:** We thank the Reviewer for this insightful question.
>
> In case we have multiple sensitive attributes, the proposed scheme can be extended in two possible ways:
>
> *Approach 1-* Applying multiple regularizers: Multiple fairness regularizers (see Eq. 3) corresponding to different sensitive attributes can be introduced and optimized jointly during the training. In fact, we do consider such an extension as a potential framework.
>
> *Approach 2-* Redefining inter/intra-edges: We can change the definition of inter- and intra-edges, where if the norm distance of multiple sensitive attributes between different nodes is lower than a certain threshold, the corresponding edges can be referred as intra-edges, while the remaining edges become inter-edges. Afterwards, we can try to balance the predicted probabilities for intra- and inter-edges with a regularizer for a better fairness.
>
> **C2:** Thank you for this comment. We agree with the reviewer and in our final submission, we will specify that this claim generally holds for social networks.
>
> **C3:** Thank you for this valuable comment. Indeed, the maximization is over k, which will be clarified better in the final submission and the corresponding equation will be corrected.
>
> **C4:** In Preliminaries, we define $\mathbf{S}$ as the one-hot encoding of sensitive attributes; and $\mathbf{e} _ {k}$ is defined in Line 168 as the unit vector whose only non-zero index is k (thanks to your comment, we have realized that there is a typo in its dimension, where it will be corrected to $\mathbb{R}^{k}$ in the final submission). The term $\mathbf{S}\mathbf{e}_{k}$ is the matrix multiplication of these two terms.
>
> **C5:** Thank you for this question.
>
> For link prediction, we had also obtained results with Common Neighbor (CN) [R3] and two-layer GCN models, where one-layer GCN led to the best AUC performance for our datasets. Thus, we chose our base GNN to be a one-layer GCN.
>
> We will further clarify this in Appendix G in our final version.
>
> **C6:** We would like to kindly mention that the datasets and sensitive attributes are also used in prior fair graph ML studies; including the baselines Adversarial, FairDrop, and FairAdj. Hence, we followed the same experimental setup in said studies without choosing the sensitive attributes by ourselves, for fair comparison. We will mention this in the final version.
>
> That said, from a purely mathematical point of view, the fairness problem considered in our study can be regarded as the problem of decorrelating the system output from a particular attribute associated with the nodes, regardless of the attribute’s relevance to real life. In this sense, the selection of sensitive attributes is not expected to affect the evaluation of such decorrelation from an algorithm design point of view.
>
> **C7:** Thank you for your keen observation, we will fix this issue in our final submission.
>
> ---
>
> [R1] E. Dai and S. Wang, “Learning fair graph neural networks with limited and private sensitive attribute information,” IEEE Transactions on Knowledge and Data Engineering, 2022.
>
> [R2] Latanya Sweeney. 2013. Discrimination in online ad delivery. Queue 11, 3 (2013), 10–29.
>
> [R3] Liben-Nowell, David, and Jon Kleinberg. "The link prediction problem for social networks." Proceedings of the twelfth international conference on Information and knowledge management. 2003.

---

> > ### Comment · Reviewer_miG4 · 2024-08-12
> >
> > Many thanks to the authors for responding to the points I raised, I think the additional clarifications make sense. The "sticking point" regarding the fairness definition is difficult to resolve in the absence of a practical study which shows that, indeed, graph ML systems do exhibit the type of bias the authors argue about, as has been shown for other types of ML techniques in a variety of deployment scenarios. Nevertheless, as I mentioned before, I do not see this as a reason to penalise this particular paper. I am retaining my original score as I think it accurately reflects my assessment of the work.

---

### Official Review · Reviewer_wVX7 · 2024-07-18

**Soundness:** 3
**Presentation:** 3
**Contribution:** 2
**Rating:** 6
**Confidence:** 3

**Summary:**

The impact of generative learning algorithms on structural bias is investigated in this paper. The authors provide a theoretical analysis on the sources of structural bias which result in disparity. Then a novel fairness regularizer is designed to alleviate the structural bias for link prediction tasks over graphs. Furthermore, a fair generation framework called FairWire is proposed by leveraging the fairness regularizer in a generative model. Finally, extensive experiments on synthetic and real datasets are conducted to show the effectiveness of mitigating structural bias in machine learning algorithms on graphs.

**Strengths:**

1. The theoretical analysis on the sources of structural bias is novel and intuitive, which provides an interesting perspective on what causes the bias problem in machine learning algorithms over graphs.
2. The design of the regularizer and fair graph generation model address the challenges and theory reasonably. Besides, sufficient experiments are provided to prove the better performance on mitigating the structural bias problem.
3. The paper's writing is good, and the overall structure is clear.

**Weaknesses:**

1. This paper aims to address the fairness in graph generation. But there are not sufficient experiments on different representative backbone generation models. The authors only provide one backbone generation model, I suggest more representative backbone generation model to solidify the effectiveness.
2. This paper provides the theory and algorithm based on the sensitive group which may limit the use to real world graphs.

**Questions:**

1. The author introduce GraphMaker as a backbone generative model and claimed that this method considers the bias mitigation when generating new graphs. Does it mean that FairWire can not been used on other generative models?
2. In the experiments, for the supervised tasks the author use link prediction tasks results, while for generation tasks, the author use node classification tasks results. What is the results of node classification tasks for supervised tasks? It would be better that the same tasks are used to evaluate both the supervised and generative tasks.
3. The authors claim the contribution in fairness over generative graph algorithm, but also provides a lot experimental results on supervised tasks. What is the difference in structural bias problem over generative tasks and supervised tasks?
4. From the experimental results, FairWire makes the performance of AUC worse. What is the influence of FairWire on the accuracy of the GNN model? Does it influence the model training or that because GNN models conducts bias on different samples?
5. From the theoretical analysis of the paper, we have to know the sensitive group first, how would you define and select sensitive groups? What kind of influence does sensitive group have on GNN and the proposed FairWire method?

**Limitations:**

The authors adequately addressed the limitations of their methods. And It is hard to foresee any potential negative societal impact of this work.

---

> ### Author Rebuttal · Authors · 2024-08-07
>
> We thank the Reviewer for their insightful comments regarding our work. We have addressed the Reviewer’s comments, and placed our responses to each comment below. Please note that references written in [R#] format are provided at the end of this rebuttal, whereas the ones in [#] format correspond to the respective references in the paper.
>
> **Q1:** Thank you for this question.
>
> The proposed fairness regularizer (Eq. 3) in FairWire can be utilized in any generative model outputting probabilistic graph topologies, including but not limited to graph autoencoder-based or random walk-based graph generation models (see Remark [*Applicability to general generative models*] in the submission). In other words, although our submission focuses on its use for graph diffusion models, the proposed framework provides a versatile use with different graph generative models.
>
> That said, to address the reviewer's comment, we also obtained results for the use of $\mathcal{L}_{\text{FairWire}}$ in a variational graph autoencoder-based (VGAE-based) graph generation [R1]. Note that VGAE does not inherently possess the ability to generate synthetic node features  and sensitive attributes. Thus, we used the real nodal features and sensitive attributes with the synthetic graph structures for evaluation. Tables below report the link prediction results, with models trained over real ($\mathcal{G}$) and synthetic graphs for Cora and Citeseer networks (see Experimental Setup in Subsection 6.1). These additional results also confirm the effectiveness of our proposed tool for fair graph generation.
>
> |  Cora | AUC (%)  | $\Delta _ {SP} $(%) | $\Delta _ {EO} $(%) |
> | :--- | :----:  |  :----:  | ---: |
> | $\mathcal{G}$ |**94.92** | 27.71 | 11.53 |
> | VGAE | 92.51| 34.99 | 9.40|
> | FairWire | 92.00|  **5.08**  |  **2.07**|
>
> |  Citeseer | AUC (%)  | $\Delta _ {SP} $(%) | $\Delta _ {EO} $(%) |
> | :--- | :----:  |  :----:  | ---: |
> | $\mathcal{G}$ |**95.76** | 29.05 | 9.53 |
> | VGAE | 92.30| 33.79 | 9.62|
> | FairWire | 92.79|  **4.05**  |  **1.88**|
>
> **Q2:** Thank you for this question.
>
> We would like to clarify that for graph generation, we present our results for both link prediction (see Table 3) and node classification (see Table 4). In addition to graph generation, we also present supervised link prediction results in Table 2 to exemplify another use of the proposed regularizer. In fact, our regularizer can be employed for any training scheme outputting edge probabilities (e.g., supervised link prediction, graph generation).
>
> **Q3:** First, we would like to clarify that our link prediction and node classifications results in Tables 3 and 4 are obtained over synthetic graphs created via generative models. In addition to them, we also wanted to demonstrate the effectiveness of the proposed regularizer for a different learning framework to show its versatile use (i.e., supervised link prediction) in Table 2.
>
> Apart from this clarification, we want to thank you for your insightful question. While it has been demonstrated that learning over real graphs for supervised tasks leads to algorithmic bias due to the structural bias in graphs [11], use of synthetic graphs makes the fairness issue more complex and challenging to resolve. As generative models learn to mimic the frequent patterns in real data (i.e., well-represented groups), there is a high probability that they will overlook the patterns exhibited within under-represented groups. Hence, they are prone to amplify the already existing structural bias [48]. We also confirmed this structural bias amplification empirically via our results in Table 1, which shows that the use of generative models worsens all fairness metrics over four different real-world datasets and motivates our proposed framework.
>
> **Q4:** In general, for all fairness-aware interventions, we expect to observe a fairness/utility trade-off [R2, R3], as bias mitigation algorithms introduce a fairness-related metric to consider in addition to utility, which typically leads to a solution that is not optimal for utility-only considerations. Thus, the observed utility/fairness trade-off is a natural outcome of introducing the proposed fairness regularizer (Eq. 3) in training to make it fairness-aware. Note that a similar trade-off can also be observed for our fairness-aware baselines (Adversarial, FairAdj, FairDrop) as well. Our results in Tables 2, 3, and 4 demonstrate that FairWire typically leads to the best utility/fairness trade-off compared to these works.
>
> **Q5:** In fairness literature, sensitive attributes are defined as the ones on which we do not want our algorithm’s predictions to depend for fair decision making. For example, recidivism predictions made by a ML model should not be related to the ethnicity of the convicts, making ethnicity the sensitive attribute and different ethnic groups the sensitive groups. Overall, the selection of sensitive groups is mainly governed by the application in which the learning model is used.
>
> In most of the fair ML literature, it is a common practice to assume that the sensitive attributes are known and given, see [7, 24, 27]. Furthermore, the real-world datasets we employed in our experiments and the sensitive groups within them have been used in existing fairness-aware graph ML works, see e.g., [11, 36, 27, 19].
>
> We hope this explanation addresses your concern.
>
> ---
> [R1] T. N. Kipf et al., “Variational graph auto-encoders,” NeurIPS Workshop on Bayesian Deep Learning, 2016.
>
> [R2] S. Dehdashtian et al., "Utility-Fairness Trade-Offs and How to Find Them," CVPR, 2024.
>
> [R3] M. Pannekoek et al., "Investigating trade-offs in utility, fairness and differential privacy in neural networks," arXiv preprint arXiv:2102.05975 (2021).

---

> > ### Comment · Reviewer_wVX7 · 2024-08-12
> >
> > Thanks for the authors satisfactorily addressing my concerns. The clarification makes sense and I have raised the presentation score and the overall score.

---

### Author Rebuttal · Authors · 2024-08-06

We would like to thank the Reviewers for their detailed reviews and constructive suggestions. We have addressed the questions raised by the Reviewers, and presented the detailed responses in a point-to-point manner.

---

### Decision · Program_Chairs · 2024-09-25

**Decision:**

Accept (poster)

**Comment:**

This paper proposes a graph generation approach with fairness consideration. The authors offer theoretical analysis to bound the representation disparity between a specific group and the remaining nodes and the statistical parity. These theoretical results further lead to the proposed FairWire that aims to push the generated graph to have a desired number of intra-/inter-edges.


All reviewers appreciate the theoretical analysis of this work and acknowledge the empirical performance of FairWire. There are concerns about the clarity in terms of math notations and experiments (e.g., visualizing results in node classification, choice of base GNN, analysis of results). The authors' responses are quite comprehensive and address the concerns well. Another concern raised by the reviewers is that the paper adopted the fairness definition without much justification. I agree that whether a fairness consideration is the appropriate or not is often a much broader topic involving many disciplines. And a more comprehensive justification from many disciplines or regulations would be more ideal, though the justification we often provide is either hypothetical from empirical results or event reported in news articles.  Though being a valid concern, I think this is relatively a minor issue given the volume of works in fair graph learning, especially on group fairness and dyadic fairness.


Overall I think this paper offers a good theoretical understanding toward group/dyadic fairness with multi-valued sensitive attribute, and I recommend acceptance.